# Alternative transcription start site selection in *Mr-OPY2* controls lifestyle transitions in the fungus *Metarhizium robertsii*

Na Guo[1], Ying Qian[1,4], Qiangqiang Zhang[1], Xiaoxuan Chen[1], Guohong Zeng[1], Xing Zhang[1], Wubing Mi[1], Chuan Xu[1], Raymond J. St. Leger[2] & Weiguo Fang[1,3]

*Metarhizium robertsii* is a versatile fungus with saprophytic, plant symbiotic and insect pathogenic lifestyle options. Here we show that *M. robertsii* mediates the saprophyte-to-insect pathogen transition through modulation of the expression of a membrane protein, Mr-OPY2. Abundant Mr-OPY2 protein initiates appressorium formation, a prerequisite for infection, whereas reduced production of Mr-OPY2 elicits saprophytic growth and conidiation. The precise regulation of Mr-OPY2 protein production is achieved via alternative transcription start sites. During saprophytic growth, a single long transcript is produced with small upstream open reading frames in its 5′ untranslated region. Increased production of Mr-OPY2 protein on host cuticle is achieved by expression of a transcript variant lacking a small upstream open reading frame that would otherwise inhibit translation of Mr-OPY2. RNA-seq and qRT-PCR analyses show that Mr-OPY2 is a negative regulator of a transcription factor that we demonstrate is necessary for appressorial formation. These findings provide insights into the mechanisms regulating fungal lifestyle transitions.

[1] Institute of Microbiology, Zhejiang University, Hangzhou 310058, China. [2] Department of Entomology, University of Maryland, College Park, MD 20742, USA. [3] Institute of Insect Sciences, Zhejiang University, Hangzhou 310058, China. [4] Present address: Chongqing Vocational College of Transportation, Chongqing 402247, China. Na Guo and Ying Qian contributed equally to this work. Correspondence and requests for materials should be addressed to W.F. (email: wfang1@zju.edu.cn)

Many fungal pathogens of plants, insects and mammals can switch between parasitic, saprophytic and symbiotic lifestyles in response to changing environmental conditions. Understanding the mechanisms by which they switch between these disparate lifestyles has important implications for agriculture and medicine[1–4]. *Metarhizium robertsii* is an excellent example of a fungus with multifactorial lifestyles[5]. It is well adapted to life in the soil as a saprophyte, and some isolates such as ARSEF2575 (Mr2575) can colonize plant roots and promote plant growth[6–9]. Mr2575 is also an entomopathogen with an ability to kill a wide spectrum of insects, and accordingly has been developed as a biocontrol agent against agricultural pests and vectors of human diseases[10].

The transition from saprophyte to pathogen is initiated when conidia adhere to the cuticle of a susceptible insect host and produce germ tubes, which differentiate into infection structures called appressoria (sticky holdfasts that attach to the cuticle). The appressoria produce infection pegs which penetrate the cuticle via a combination of mechanical pressure and cuticle-degrading enzymes. The fungus proliferates in the host hemocoel as a yeast-like phase (blastospores), and the insect is killed by a combination of fungal growth and toxins. Hyphae subsequently re-emerge from the cadaver to produce conidia that have the potential to enter into saprophytic, symbiotic or pathogenic lifestyles[11].

Many genes involved in the lifestyle switch to pathogen have been experimentally characterized; their gene products are generally expressed at low levels outside the host. They include an adhesin (MAD1) and hydrophobins that are responsible for adherence to the cuticle[12,13], two chitin synthases for appressorial formation[14], and a large number of cuticle-degrading enzymes for penetration of the cuticle[15,16]. Gene products associated with colonizing the hemocoel include the cold shock protein CRP1, laccase Mlac1, sterol carrier Mr-NPC2a, the collagen-like protein MCL1, enzymes for anaerobic respiration, and toxic secondary

metabolites such as destruxins[17–22]. Major signaling pathway MAPK cascades and cAMP-PKA have been found to regulate both saprophytic growth and pathogenesis[23,24]. Nevertheless, no mechanisms have been characterized that control *M. robertsii's* choice of lifestyle options as a saprophyte or a pathogen.

Here, from analysis of a random T-DNA insertion library[22], we identify a membrane anchor protein (Mr-OPY2) that controls the saprophyte-to-pathogen transition of *M. robertsii*. Mr-OPY2 protein levels are low during saprophytic growth, and when elevated they initiate appressorial formation. Precise regulation of the Mr-OPY2 protein level is achieved via alternative transcription start sites. We further find that Mr-OPY2 controls appressorial formation by regulating a previously unidentified transcription factor, AFTF1 (appressorial formation transcription factor 1).

## Results

**Identification of the membrane anchor protein Mr-OPY2.** From a T-DNA insertion mutant library, we identified a mutant (M2880) that cannot infect insects. Gene mapping showed that the T-DNA in the mutant was inserted in the open reading frame (ORF) of a gene (MAA_03000) that encodes a protein with significant similarity ($1e^{-08}$) to the membrane anchor protein OPY2p from *Saccharomyces cerevisiae* (AJV94457), and we designated this gene as *Mr-OPY2* (*M. robertsii* OPY2). *Mr-OPY2* is a single copy gene with an 1,305 bp ORF that is interrupted by one intron, and encodes a protein containing 434 amino-acid residues. The deduced Mr-OPY2 has an OPY2 domain (PFAM09463) from amino acids 30 to 68, and a transmembrane domain from amino acids 93–115 as predicted by TMHMM (version 2.0)[25], and domains showing significant similarity to CR-A and D in *S. cerevisiae* Opy2p[26]. Blast analysis found that homologs of Mr-OPY2 were widely distributed in both

**Table 1 Transcripts and strains in this study**

| Name | Description | Ref |
|---|---|---|
| **Transcripts** | | |
| Mr-OPY2-S | The short transcript of the Mr-OPY2 gene | This study |
| 5′ UTR$^S$ | The 5′UTR of the short transcript Mr-OPY2-S | |
| Mr-OPY2-L | The long transcript of the Mr-OPY2 gene | This study |
| 5′ UTR$^L$ | The 5′UTR of the long transcript Mr-OPY2-L | |
| Mr-OPY2-L$^{\Delta AUGs}$ | The mutant of Mr-OPY2-L with the 5′UTR$^L$ uORFs mutated | This study |
| **Genomic clones** | | |
| gMr-OPY2 | The genomic clone of the Mr-OPY2 gene | This study |
| gMr-OPY2$^{\Delta AUGs}$ | The mutant of gMr-OPY2 with the 5′UTR uORFs mutated | This study |
| **Strains** | | |
| WT | The wild-type strain of M. robertsii ARSEF2575 | |
| M2880 | A mutant with T-DNA inserted in the ORF of the Mr-OPY2 gene | This study |
| ΔMr-OPY2 | The mutant with Mr-OPY2 ORF deleted based on homologous recombination | This study |
| C−ΔMr-OPY2 | The complemented strain of ΔMr-OPY2 | This study |
| ΔMr-STE50 | The mutant with Mr-STE50 ORF deleted based on homologous recombination | This study |
| C-ΔMr-STE50 | The complemented strain of ΔMr-STE50 | This study |
| T-Mr-OPY2-LΔ$^{AUGs}$ | The strain with gMr-OPY2Δ$^{AUGs}$ to replace its corresponding region in M. robertsii | This study |
| T-Mr-OPY2-L$^{AUG}$ | The strain with gMr-OPY2 to replace its corresponding region in M. robertsii | This study |
| ΔMr-OPY2::Mr-OPY2-S | The strain with Mr-OPY2-S controlled by the constitutive promoter Pgpd | This study |
| ΔMr-OPY2::Mr-OPY2-L | The strain with Mr-OPY2-L controlled by the promoter Pgpd from M. acridum | This study |
| ΔMr-OPY2:: Mr-OPY2-LΔ$^{AUGs}$ | The strain witht Mr-OPY2-LΔ$^{AUGs}$ controlled by the Pgpd from M. acridum | This study |
| P404-Mr-OPY2 | The strain with Mr-OPY2 mORF driven by the 404 bp Pgpd from M. acridum | This study |
| P683-Mr-OPY2 | The strain with Mr-OPY2 mORF driven by the 683 bp Pgpd from M. acridum | This study |
| ΔMero-Fus3 | The deletion mutant of the Fus3-MAPK gene | Ref. 24 |
| ΔMero-Hog1 | The deletion mutant of the Hog1-MAPK gene | Ref. 24 |
| ΔAftf1 | The deletion mutant of the Aftf1 gene | This study |
| C-ΔAftf1 | The complemented strain of ΔAftf1 | This study |
| Aftf1$^{OE}$ | The strain overexpressing the Aftf1 gene | This study |

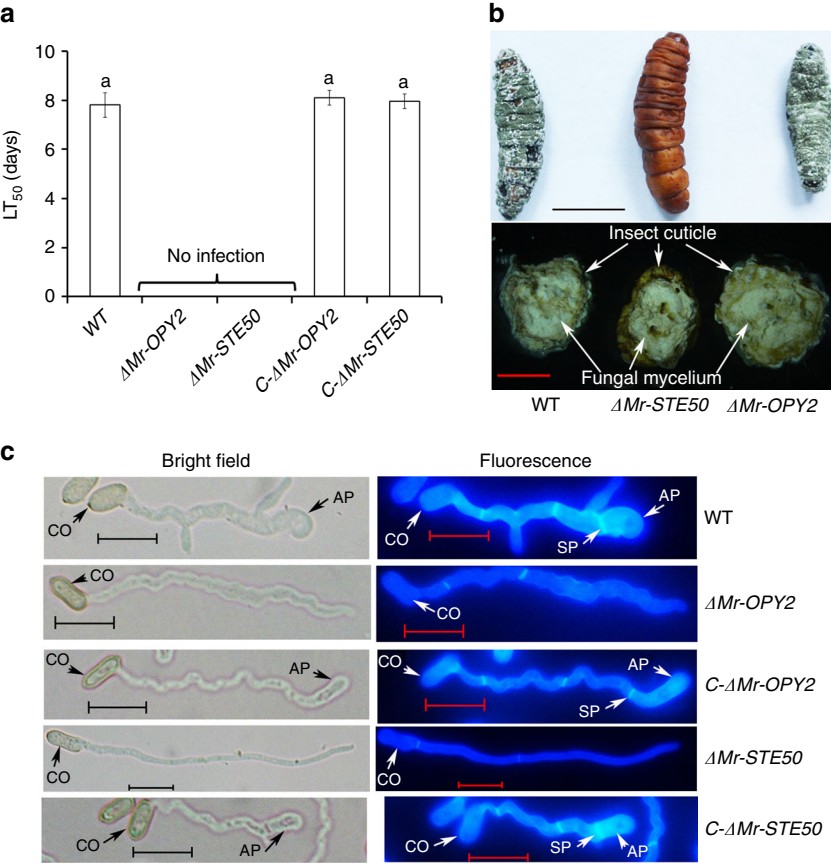

**Fig. 1** Pathogenicity of WT, the mutants *ΔMr-OPY2* and *ΔMr-STE50*, and their respective complementation strains. **a** LT50 (time taken to kill 50% of insects) values when the insects were inoculated by topical application. The bioassays were repeated three times with 40 insects per repeat. Data are expressed as the mean ± SE. Values with different letters are significantly different ($n = 3$, $P < 0.05$, Tukey's test in one-way ANOVA). **b** Upper panel: mycelial growth and conidiation on the surface of cadavers (scale bar represents 10 mm). Lower panel: mycelia in cross sections of cadavers (scale bar represents 5 mm). Each image is representative of ~120 insect cadavers (three replicates with 40 insects per replicate). **c** Formation of appressoria (stained with Calcofluor Brightener White 2B) against a hydrophobic plastic surface. AP: appressorium; CO: conidium; SP: septum. Left: bright field microscopy. Note: the hyphal tips of *ΔMr-OPY2* and *ΔMr-STE50* do not swell to form appressoria. Right: fluorescence microscopy. Note: the septum between the appressorium and its appressorial mother cell. Images are representative of at least three independent experiments for each condition. Scale bar represents 10 μm

pathogenic and saprophytic Ascomycota and Basidiomycota. Phylogenetic analysis showed that the clade containing OPY2 proteins from Ascomycota yeasts was basal to the clade that contained OPY2 proteins of filamentous Ascomycota and Basidiomycota. The OPY2 genes of the filamentous fungi conform to their species tree, consistent with their being derived from a common ancestral sequence (Supplementary Fig. 1a, Supplementary Table 1). We compared the topology of the obtained tree with those of alternative trees using nine tests including the SH-test and the AU-test provided by the program CONSEL[27]. The nine tests consistently showed that the obtained tree (Supplementary Fig. 1a) to be the best supported (Supplementary Fig. 1b).

**Mr-OPY2 and pathogenicity.** We disrupted *Mr-OPY2* in the wild-type strain (WT) to produce *ΔMr-OPY2*. The genes and strains used in this study are listed in Table 1. *ΔMr-OPY2* was complemented with its genomic clone containing the upstream (1,747 bp) and downstream (245 bp) regions to produce strain *C-ΔMr-OPY2*. The confirmation of gene disruption and mutant complementation is presented in Supplementary Fig. 2.

The pathogenicity of *M. robertsii* was assayed on *Galleria mellonella* larvae. Inoculations were conducted either by topically applying conidia onto the insect cuticle or by direct injection of conidia into the hemocoel (thus bypassing the cuticle). As with the T-DNA insertion mutant M2880, *ΔMr-OPY2* was unable to infect insects via topical application (Fig. 1a), but it was as pathogenic as the WT following injection (Supplementary Fig. 3). *C-ΔMr-OPY2* showed WT levels of virulence (Supplementary Fig. 3) following topical application or direct injection (Fig. 1a and Supplementary Fig. 3). Insects killed by either the WT or injected *ΔMr-OPY2* were mummified with dark green conidia produced on the cadavers (Fig. 1b).

We investigated whether the mutants were able to produce appressoria against insect cuticle and another normally inductive milieu-the hydrophobic surface of plastic petri dishes in the presence of low levels of nitrogenous nutrients[28]. *ΔMr-OPY2* germ tubes meandered across the plastic and hindwings of *Locusta migratoria manilensis*, but their hyphal tips did not swell to initiate appressorial formation. Fluorescent staining with Calcofluor white Brightener 2B showed that the fluorescent intensity of the *ΔMr-OPY2* hyphal tip was not distinguishable from other parts of the hyphae (Fig. 1c), whereas both WT and *C-ΔMr-OPY2* produce normal appressoria that fluoresce strongly (Fig. 1c). Therefore, the cell wall structure or composition is altered in the growing hyphal tips of *ΔMr-OPY2*. In addition, appressorial differentiation of *M. robertsii* typically occurs after nuclear division with a septum being formed between the

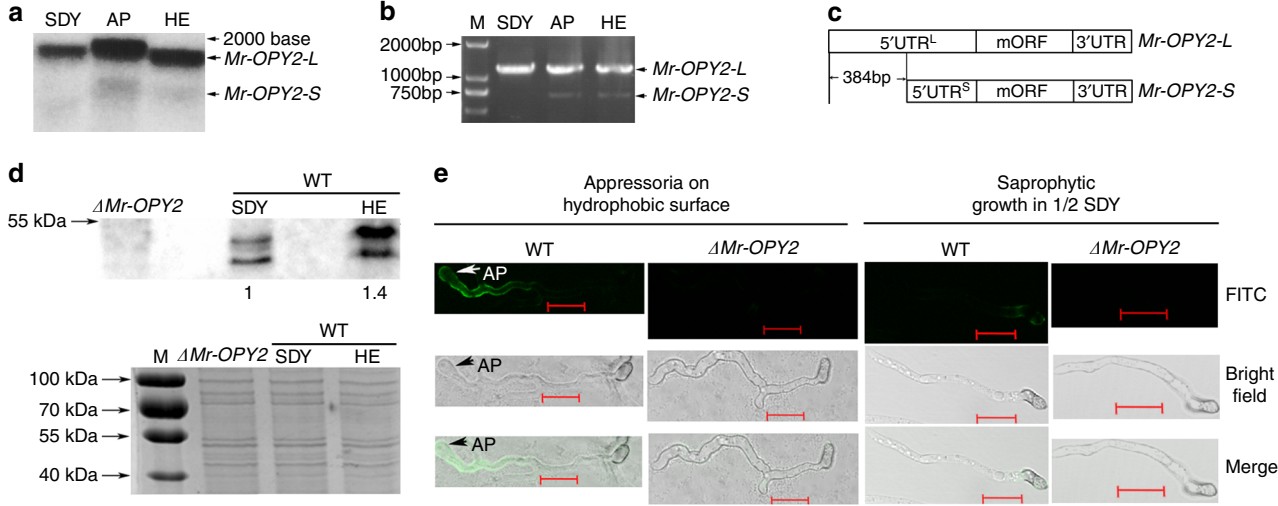

**Fig. 2** Transcription and translation analyses of the *Mr-OPY2* gene. **a** Representative Northern blot image of RNA isolated from undifferentiated mycelia grown in SDY or hemolymph (HE), and from germlings differentiating appressoria on locust wings (AP). The uncropped Northern blot is shown in Supplementary Fig. 11a. **b** 5′RACE with the RLM RT-PCR kit. The uncropped image of the agarose gel is shown in Supplementary Fig. 11b. **c** A diagrammatic representation of differences between the two mRNA variants of *Mr-OPY2*. Note: the two transcripts share the same major ORF but they have different 5′ UTR. **d** Representative western blot image and analysis to quantify Mr-OPY2 levels in Δ*Mr-OPY2* and WT grown in SDY or hemolymph (HE) (Upper panel). The OPY2 levels of hemolymph-grown WT mycelium (1.4) is calculated relative to growth in SDY which is set at 1. Δ*Mr-OPY2* is included as a negative control. Lower panel: a portion of the loading control gel (for the Western blot analysis in upper panel) stained with Coomassie Brilliant Blue. M: Protein ladder (Thermo scientific, USA). The uncropped images of the western blot and the SDS-PAGE gel are shown in Supplementary Fig. 11c, d. **e** Immunohistochemical staining of the Mr-OPY2 protein in germlings differentiating appressoria against a plastic surface, and in non-differentiating germlings from 1/2 SDY shake cultures. Scale bar represents 10 μm. FITC: Fluorescein isothiocyanate; Merge: FITC and bright field pictures are merged; AP: appressorium. Note: no fluorescent signal in Δ*Mr-OPY* and a weak signal in WT grown in 1/2 SDY. Images are representative of at least three independent experiments for each condition

appressorium and appressorial mother cell[28]. This septation was observed in WT and *C-ΔMr-OPY2*, but not in Δ*Mr-OPY2* (Fig. 1c). WT and *C-ΔMr-OPY2* germlings typically had three septa when the hyphal tips initiated appressorial formation, at which time Δ*Mr-OPY2* only had two. 18 h after inoculation, when ∼ 80% of WT hyphal tips had produced appressoria against plastic surfaces, the hyphae of WT (27.4 ± 1.82 μm) were significantly (*t*-test, *P* = 0.007) shorter than Δ*Mr-OPY2* (42.7 ± 0.29 μm), as Δ*Mr-OPY2* showed continued polar growth rather than differentiation into appressoria.

**Mr-OPY2 has two mRNA variants during infection.** Northern blots and 5′ and 3′ RACE (Rapid Amplification of cDNA Ends) were used to analyze RNA during: (1) saprophytic growth (in SDY broth (Sabouraud dextrose broth plus 1% yeast extract)); (2) cuticle penetration (conidia germinating and differentiating appressoria on locust wings) and 3) hemocoel colonization (hyphae cultured in *Bombyx mori* (silkworm) hemolymph). Northern analysis revealed that a single *Mr-OPY2* transcript was produced during saprophytic growth, and an additional less intense smaller band was produced in the two infection stages (Fig. 2a). RACE was performed using the RLM RT-PCR kit (Roche, USA) that only clones intact mRNA with a 7-methyl guanosine cap structure. Only one PCR band was obtained from the three RNA samples with 3′ RACE. Consistent with Northern analysis, 5′ RACE obtained one long band using RNA from *M. robertsii* mycelia grown in SDY broth, and an additional shorter band from *M. robertsii* when penetrating cuticle or growing in hemolymph (Fig. 2b). All PCR products were cloned, sequenced and assembled. The long 1,836 bp and shorter 1,452 bp transcripts are designated as *Mr-OPY2-L* (Genbank number: KY548479), and *Mr-OPY2-S* (Genbank number: KY548480), respectively. The two mRNA variants contain an identical ORF

(designated as major ORF:mORF) that encodes Mr-OPY2 (Fig. 2c), and differ only in that the 5′ UTR of *Mr-OPY2-L* (designated as 5′ UTR^L) is 384 bp longer than that of *Mr-OPY2-S* (designated as 5′ UTR^S). The first nucleotide of *Mr-OPY2-L* (T) is different from that of *Mr-OPY2-S* (C), and no intron was found in the region corresponding to the UTRs, indicating that the two mRNA variants result from alternative transcription start sites.

Western blot analysis showed that there was 1.4-fold more Mr-OPY2 protein in hemolymph cultures than in SDY (Fig. 2d). We could not obtain sufficient biomass from differentiating appressoria to extract membrane proteins for western analysis, so we used indirect immunofluoresence (IIF) assays to compare Mr-OPY2 protein levels in appressoria with those in non-differentiating germlings in SDY. The assay results showed that the Mr-OPY2 protein was more abundant in appressoria (Fig. 2e).

To identify differences in copy number between *Mr-OPY2-L* and *Mr-OPY2-S* we used two pairs of primers for quantitative RT-PCR (qRT-PCR) analysis of the 5′ UTR^L and mORF (Supplementary Fig. 4). No significant differences were found in the levels of 5′ UTR^L and mORF during saprophytic growth and the two infection stages. Therefore, expression of the *Mr-OPY2-S* transcript did not significantly contribute to mORF copy number. Consequently, changes in Mr-OPY2 protein levels during lifestyle transitions are most probably attributable to translational regulation of mORF in *Mr-OPY2-L* and *Mr-OPY2-S*.

**uORFs mediate translational suppression of mORF in Mr-OPY2-L.** The 5′ UTR^S of *Mr-OPY2-S* lacks uORFs (upstream ORFs), whereas two uORFs were identified in the 5′ UTR^L of *Mr-OPY2-L* (Fig. 3a). As uORFs suppress translation efficiency in many genes[29–31], we investigated the effect of the uORFs on translation of the main ORF in *Mr-OPY2-L*. To do this, we cloned

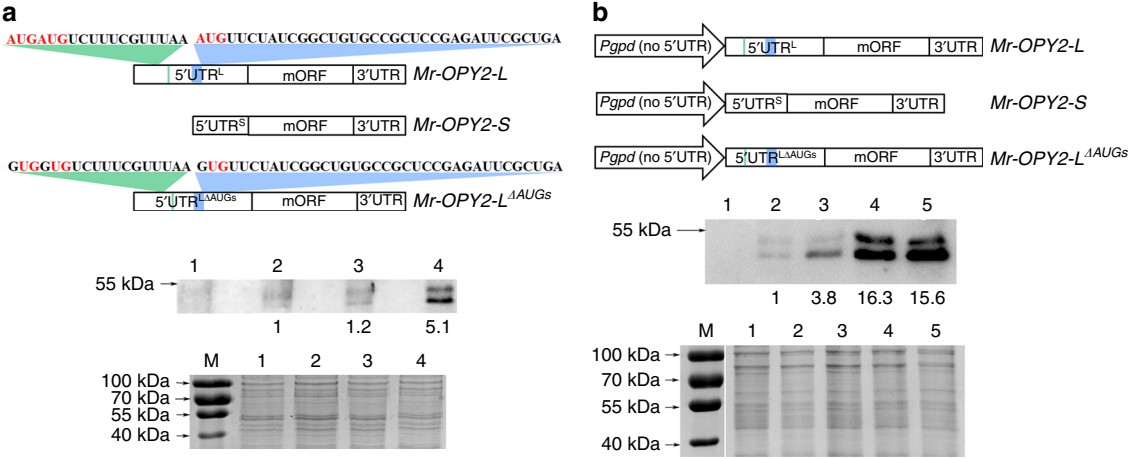

**Fig. 3** The uORFs suppress translation efficiency of the major ORF in *Mr-OPY2-L*. **a** Top panel: graphic representation of WT transcripts *Mr-OPY2-L*, *Mr-OPY2-S* and the mutated *Mr-OPY2-L* (*Mr-OPY2-L$^{\Delta AUGs}$*) that has GUGs in place of AUGs (highlighted in red). The blue and green labelled sequences in the 5′UTR represent the positions of the two different uORFs (these color conventions are also used in the top panel in **b**. Middle panel: Mr-OPY2 protein levels in SDY grown cultures of: (1) Δ*Mr-OPY2*; (2) WT; (3) *T-Mr-OPY2-L$^{AUG}$* (the native *Mr-OPY2* gene was replaced by a native genomic *Mr-OPY2* clone); and (4) *T-Mr-OPY2$^{\Delta AUGs}$* (the native *Mr-OPY2* gene was replaced by a genomic *Mr-OPY2* clone with mutated uORFs). Bottom panel: a portion of the loading control gel (for the Western blot analysis in the middle panel) stained with Coomassie Brilliant Blue. M: Protein ladder. **b** The short *Mr-OPY2-S* transcript is more efficiently translated than the longer *Mr-OPY2-L*. Top panel: graphic representation of the constructs containing the transcript *Mr-OPY2-L*, *Mr-OPY2-S*, and *Mr-OPY2-L$^{\Delta AUGs}$* driven by the *Pgpd-NUTR*. The 'No 5′UTR' indicates that the 5′UTR of the *M. acridum gpd* gene is excluded. These constructs were transformed into the *Mr-OPY2* deletion mutant Δ*Mr-OPY2*. Middle panel: representative Western blot image and analysis to quantify Mr-OPY2 levels in SDY grown cultures of: (1) Δ*Mr-OPY2*; (2) WT; (3) Δ*Mr-OPY2::Mr-OPY2-L*; (4) Δ*Mr-OPY2::Mr-OPY2-S*, and (5) Δ*Mr-OPY2::Mr-OPY2-L$^{\Delta AUGs}$*. Bottom panel: a portion of the loading control gel (for the western blot analysis in the middle panel) stained with Coomassie Brilliant Blue. M: Protein ladder. For western blot analysis (**a**, **b** middle panels), the Mr-OPY2 levels are calculated relative to the WT which is set at 1 (Δ*Mr-OPY2* is a negative control). The uncropped images of the western blots and the SDS-PAGE gels are shown in Supplementary Fig. 12. Images are representative of at least three independent experiments

the genomic DNA fragment (g*Mr-OPY2*) that contains the two exons, the single intron and the termination region (200 bp), and mutated the AUGs to GUGs in the 5′ UTR$^L$ to generate g*Mr-OPY2$^{\Delta AUGs}$* (Fig. 3a, Supplementary Fig. 5a). The knock-in method was used to replace the corresponding region in *M. robertsii*'s genome with g*Mr-OPY2$^{\Delta AUGs}$*, producing the strain *T-Mr-OPY2-L$^{\Delta AUGs}$*. As a control, the corresponding region was also replaced with the unmutated g*Mr-OPY2* to produce the strain *T-Mr-OPY2$^{AUG}$* (Supplementary Fig. 5a, b). qRT-PCR of SDY mycelial cultures revealed that expression of the *Mr-OPY2* ORF was similar ($P > 0.05$, Tukey's test in one-way analysis of variance (ANOVA)) in WT, *T-Mr-OPY2$^{AUG}$* and *T-Mr-OPY2$^{\Delta AUGs}$* (Supplementary Fig. 5c), whereas western analysis showed that Mr-OPY2 protein levels in *T-Mr-OPY2$^{\Delta AUGs}$* were increased ~4-fold compared with the WT and *T-Mr-OPY2$^{AUG}$* (Fig. 3a). No obvious difference in the level of Mr-OPY2 protein was observed between WT and *T-Mr-OPY2-L$^{AUG}$* (Fig. 3a). These results show that upstream AUG triplets in the 5′ UTR of the long mRNA variant of *Mr-OPY2* reduce translatability and protein levels.

Secondary structures of 5′ UTRs may affect translation of the downstream ORF. We thus predicted the secondary structures of 5′ UTR$^L$ and 5′ UTR$^S$ using the Sfold Web Server (http://sfold. wadsworth.org/cgi-bin/index.pl). The 5′ UTR$^L$ formed a complex secondary structure with many stem-loops, and had a Gibbs free energy $\Delta G^0_{37} = -25.6$ Kcal mol$^{-1}$. In contrast, 5′ UTR$^S$ formed a simple stem-loop structure with $\Delta G^0_{37} = -7.2$ Kcal mol$^{-1}$. The secondary structure of the 5′ UTR of *Mr-OPY2-L$^{\Delta AUGs}$* ($\Delta G^0_{37} = -26.9$ Kcal/mol) was similar to that of *Mr-OPY2-L* (Supplementary Fig. 5d). Secondary structures with free energy higher than $\Delta G = -30$ Kcal/mol are not considered to impair translation[32]. Therefore, it is unlikely that the secondary structures of 5′ UTR$^L$ and 5′ UTR$^S$ substantially effect translation of the downstream ORF.

To characterize the effects of 5′ UTR$^L$ and 5′ UTR$^S$ further, we constructed expression plasmids where the transcripts *Mr-OPY2-S*, *Mr-OPY2-L* and *Mr-OPY2-L$^{\Delta AUGs}$* were placed downstream of the constitutive *gpd* gene promoter (*Pgpd*) from *M. acridum*[33]. The 5′ UTR of the *gpd* gene was not included in the promoter to avoid it impacting translation (Fig. 3b). These plasmids were then transferred into Δ*Mr-OPY2* (i.e., the native mORF was deleted) to construct strains Δ*Mr-OPY2::Mr-OPY2-S*, Δ*Mr-OPY2::Mr-OPY2-L*, and Δ*Mr-OPY2:: Mr-OPY2-L$^{\Delta AUGs}$*. Strains with similar levels of the mORF transcript were selected for further analysis (Supplementary Fig. 6). In SDY cultures, the level of Mr-OPY2 protein in Δ*Mr-OPY2::Mr-OPY2-S* was ~4-fold higher than Δ*Mr-OPY2::Mr-OPY2-L*, indicating that *Mr-OPY2-S* was more efficiently translated than *Mr-OPY2-L*. Conversely, Δ*Mr-OPY2:: Mr-OPY2-L$^{\Delta AUGs}$* and Δ*Mr-OPY2::Mr-OPY2-S* had similar Mr-OPY2 protein levels (Fig. 3b). Taken together these results suggest that: (1) the 5′ UTR$^L$ secondary structure does not affect the translation efficiency of mORF, and (2) uORFs are major factors for suppressing the translation of mORF.

**Differences in Mr-OPY2 levels lead to lifestyle transitions.** As described above, alternative transcription start sites produce differences in Mr-OPY2 protein levels that characterize saprophytic and pathogenic lifestyles. In order to investigate the impact of Mr-OPY2 protein abundance on lifestyle transitions, we took advantage of a series of truncated *Pgpd* promoters of the *M. acridum gpd* gene which express foreign genes at different strengths[33]. We selected the 404 bp (P404) and 683 bp (P683) long promoters, which include the 5′UTR of the *M. acridum gpd* gene, to drive the major ORF of *Mr-OPY2* in the *Mr-OPY2* deletion mutant (Δ*Mr-OPY2*) (Fig. 4a), i.e., the expression of *Mr-OPY2* was exclusively controlled by the constitutive *gpd* promoters. As expected, qRT-PCR analysis showed that expression

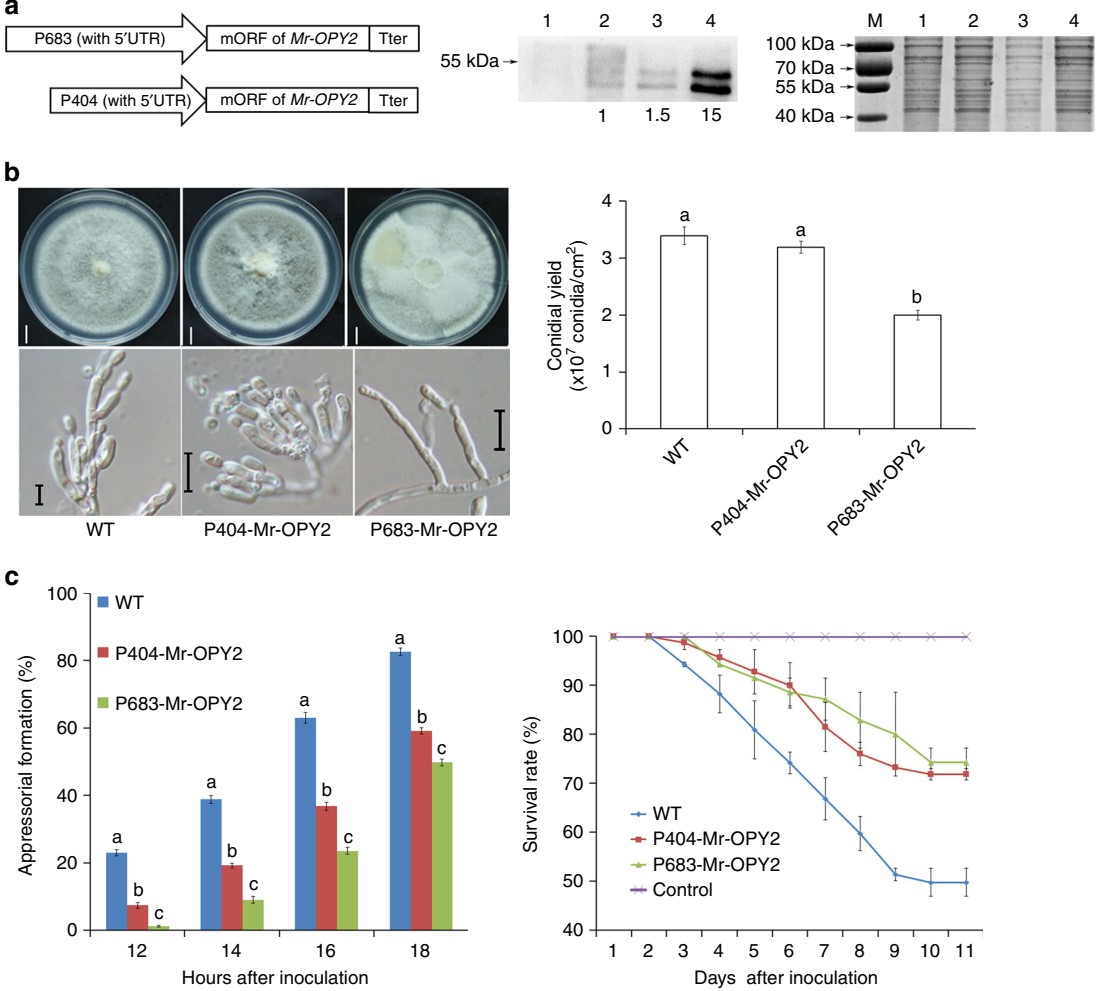

**Fig. 4** Precise regulation of Mr-OPY2 levels is important for saprophytic growth and infection. **a** Constructing strains with manipulated Mr-OPY2 levels. Left panel: diagram illustrating constructs where *Mr-OPY2's* ORF is driven by the two *Pgpd* promoters, P683 and P404, with high and low activity, respectively. These constructs were transformed into Δ*Mr-OPY2* to produce strains where the Mr-OPY2 level was exclusively controlled by the promoters. The 'With 5′UTR' indicates the 5′UTR of the *M. acridum gpd* gene is included. Middle panel: western blot analysis of Mr-OPY2 levels in the transformants: (1) Δ*Mr-OPY2*; (2) WT; (3) *P404-Mr-OPY2;* and (4) *P683-Mr-OPY2*. Mr-OPY2 levels are calculated relative to the WT which is set at 1. Δ*Mr-OPY2* is a negative control. Right panel: a portion of the loading control gel (for the Western blot analysis in the middle panel) stained with Coomassie Brilliant Blue. M: Protein ladder. The uncropped images of the western blot and the SDS-PAGE gel are shown in Supplementary Fig. 13. **b** Colony phenotype (left upper panel), conidiophores (left lower panel) and conidial yields (right panel) of strains with different Mr-OPY2 levels. Note: strain *P683-Mr-OPY2* with elevated Mr-OPY2 level relative to WT has a fluffy colony with impaired conidiophores and reduced conidial yield. Colony pictures were taken 18 days after inoculation. Scale bar in the left upper panel represents 10 mm, left lower panel 10 μm. In the right panel, values with different letters are significantly different (*n* = 9, *P* < 0.05, Tukey's test in one-way ANOVA). **c** Pathogenicity of strains with different Mr-OPY2 levels. Left panel: appressorial development on a hydrophobic surface. At each time point, values with different letters are significantly different (*n* = 3, *P* < 0.05, Tukey's test in one-way ANOVA). Right panel: survival curves of *G. mellonella* larvae infected by the WT, *P404-Mr-OPY2* and *P683-Mr-OPY2*. Control: insects treated with 0.01% Triton X-100 solution. Bioassays were repeated three times with 40 insects per repeat. Conidial yield and appressorium formation assays were repeated three times with three replicates per repeat. Data are expressed as the mean ± SE. Images are representative of at least three independent experiments for each condition

of the *Mr-OPY2* transcript in *P683-Mr-OPY2* (driven by the promoter P683) is significantly higher (*t*-test, *P* = 0.006) than expression driven by P404 (strain designated *P404-Mr-OPY2*), and the quantity of Mr-OPY2 protein in *P683-Mr-OPY2* was ∼ 8-fold higher than in either *P404-Mr-OPY2* or WT (Fig. 4a). Three isolates were randomly selected from *P404-Mr-OPY2* and *P683-Mr-OPY2* colonies. As no intra-strain differences were found between the three isolates of each strain in any assay we provide representative data for a single isolate/strain.

During saprophytic growth on PDA (potato dextrose agar) plates, the growth rate and conidial yield of *P404-Mr-OPY2* was similar to WT (Fig. 4b). However, the more highly expressing

*P683-Mr-OPY2* produced a fluffy phenotype, abnormal conidiophores and significantly (*P* < 0.01, Tukey's test in one-way ANOVA) lower conidial yield than WT (Fig. 4b). Compared with WT, appressorial formation by *P404-Mr-OPY2* and *P683-Mr-OPY2* was delayed (Fig. 4c), and the pathogenicity of both *P683-Mr-OPY2* and *P404-Mr-OPY2* was significantly (*P* < 0.05, Tukey's test in one-way ANOVA) reduced (Fig. 4c).

**Mr-OPY2 and tolerance to abiotic stresses.** As *S. cerevisiae* Opy2p is involved in tolerance to high osmotic stress (Yamamoto et al., 2016), we investigated whether *M. robertsii* OPY2 is also

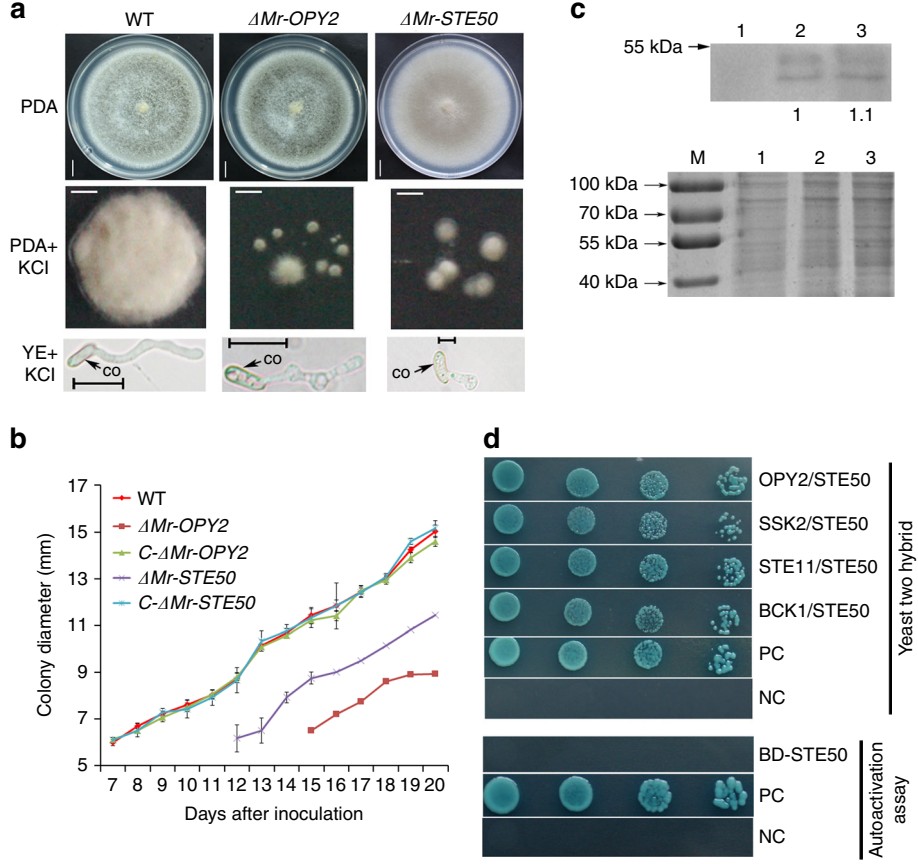

**Fig. 5** Colony phenotypes and osmotic stress tolerance of *ΔMr-OPY2*, *ΔMr-STE50* and WT. **a** Upper panel: colonies of the strains on PDA (Scale bar represents 10 mm). Middle panel: colonies on PDA plus 0.75 M KCl (scale bar represents 1 mm). Lower panel: bright field images of individual germlings in YE (0.01% yeast extract) plus 0.75 M KCl (Scale bar represents 10 µm). Colony pictures were taken 18 days after inoculation. Germling pictures were taken 16 h after inoculation. **b** Growth curves of the strains on PDA plates plus 0.75 M KCl. Growth assays were repeated three times with three replicates per repeat. Data are expressed as the mean ± SE. Note: growth of *ΔMr-OPY2* and *ΔMr-STE50* was greatly reduced. **c** Upper panel: Western blot analysis of Mr-OPY2 protein levels in *ΔMr-OPY2* grown in SDY (1), WT grown in SDY (2) and in SDY plus 0.75 M KCl (3). The band intensity of WT in SDY is set at 1, and WT in SDY with KCl is relative to it; *ΔMr-OPY2* is a negative control. Lower panel: a portion of the loading control gel (for the western blot analysis in the upper panel) stained with Coomassie Brilliant Blue. M: Protein ladder. The uncropped images of the Western blot and the SDS-PAGE gel are shown in Supplementary Fig. 13. **d** Upper panel: yeast two-hybrid confirmation of the physical interaction of Mr-SET50 with Mr-OPY2 and three MAPKKK proteins (SSK2, STE11 and BCK1). Colonies were grown in SD-His-Ade-Leu-Trp + X-α-gal + AbA (Takara, Dalian China). NC: negative control (yeast cells containing the plasmid pGADT7-T and pGBKT7-Lam). PC: positive control (yeast cells containing the plasmid pGADT7-T and pGBKT7-53). Lower panel: Mr-STE50 lacks autoactivation activity. Y2HGold cells with pGBKT7-Mr-STE50 cannot grow in SD-His-Trp-Ade + X-α-gal (Takara, Dalian China). NC and PC are the same as those in the upper panel. Images are representative of at least three independent experiments for each condition

involved in tolerance to abiotic stresses. *ΔMr-OPY2* germinates and grows in SDY broth or on PDA plates at the same rate as WT and *C-ΔMr-OPY2* (Fig. 5a, Supplementary Fig. 7), and this was not altered by adding 0.01% $H_2O_2$ to induce oxidative stress (Supplementary Fig. 7). The cell wall-disturbing agent Congo red (1 mg ml$^{-1}$) produced no significant ($P > 0.05$, Tukey's test in One-way ANOVA) differences in growth rate between WT, *ΔMr-OPY2* and *C-ΔMr-OPY2* on PDA (Supplementary Fig. 7), but the germination rate of *ΔMr-OPY2* in SDY was significantly reduced ($P < 0.05$, Tukey's test in one-way ANOVA) (Supplementary Fig. 7). Under hyperosmotic stress (SDY plus 0.75 M KCl), WT, *ΔMr-OPY2* and *C-ΔMr-OPY2* showed similar ($P > 0.05$, Tukey's test in One-way ANOVA) germination rates, but the *ΔMr-OPY2* hyphae were severely deformed forming a pearl-like shape (Fig. 5a). Similarly, unlike the *C-ΔMr-OPY2* and WT, mycelial growth of *ΔMr-OPY2* on PDA plates supplemented with 0.75 M KCl was also greatly ($P < 0.01$, Tukey's test in One-way ANOVA) inhibited (Fig. 5b). However, the level of Mr-OPY2 protein in WT was the same in SDY and in SDY supplemented with 0.75 M KCl (Fig. 5c). No significant differences were observed between WT,

*P683-Mr-OPY2* and *P404-Mr-OPY2* in their tolerance to oxidative stress, high osmotic stress and Congo red (Supplementary Fig. 8).

**Mr-OPY2 controls phosphorylation levels of MAPKs.** The *S. cerevisiae* OPY2 protein regulates the phosphorylation level of MAPKs by complexing with the adaptor protein STE50p that in turn interacts with three MAPKKK proteins[26]. We used yeast two-hybrid assays to confirm that the *M. robertsii* Mr-OPY2 physically interconnects with MAA_02467, the *M. robertsii* ortholog of STE50p (designated Mr-STE50), and thereafter the three MAPKKK proteins Mero-Ste11, Mero-Ssk2 and Mero-Bck1 (Fig. 5d). We disrupted *Mr-STE50* to produce *ΔMr- STE50*, which was complemented with its genomic clone to produce *C-ΔMr-STE50*. Confirmation of gene disruption and mutant complementation is shown in Supplementary Fig. 2. *ΔMr-STE50* produced greyish green conidia compared with the darker green conidia of WT colonies (Fig. 5a). *ΔMr- STE50* had reduced tolerance to high osmotic stress (Fig. 5b) and the cell wall-disturbing agent Congo Red (Supplementary Fig. 7), but its tolerance to

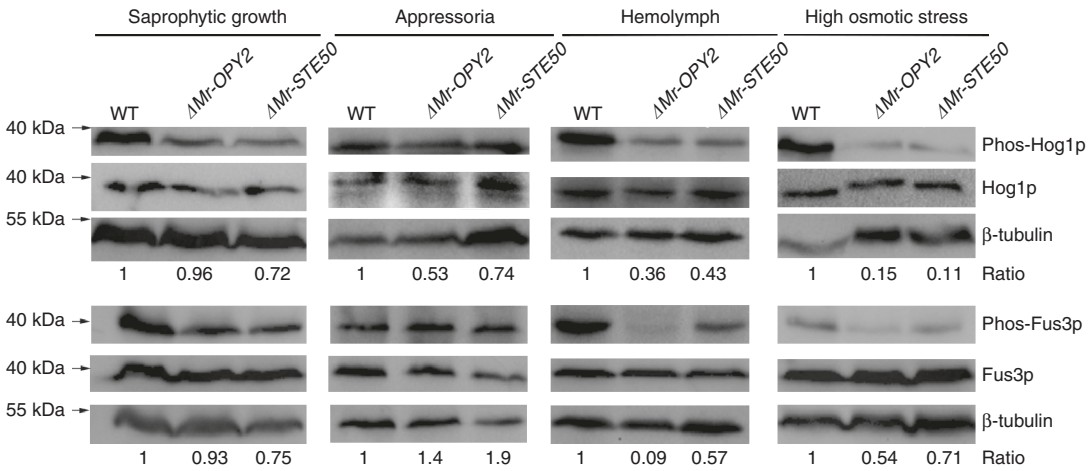

**Fig. 6** Regulation of MAPK signaling pathways by Mr-OPY2 and Mr-STE50 during saprophytic growth, pathogenesis and under high osmotic stress. β-Tubulin was used as a loading control. Numbers indicate band intensity for Phos-Fus3p (Phos-Hog1p) relative to total Fus3p (Hog1p). WT values were set to 1. Images are representative of at least three independent experiments for each condition. The uncropped images of the western blots are shown in Supplementary Fig. 14

oxidative stress was unaltered (Supplementary Fig. 7). Like ∆Mr-OPY2, ∆Mr-STE50 did not produce appressoria, and was consequently unable to kill insects via topical application (Fig. 1a, c), but it showed WT virulence when injected into insects (Supplementary Fig. 3). Mycelia of WT and ∆Mr-OPY2 grew out of cadavers commencing 3 days after death. In contrast, no mycelium emerged on the ∆Mr-Ste50 infected insects, although cadavers contained mycelia (Fig. 1b). There were no significant (P > 0.05, Tukey's test in one-way ANOVA) differences between the WT and C-∆Mr-STE50 in any assays.

Previously, we found that the Fus3-MAPK and Hog1-MAPK signaling pathways regulate M. robertsii's infectivity and tolerance to high osmotic stress[24]. To investigate if Mr-OPY2 regulates these two MAPK signaling pathways we compared the phosphorylation level of Fus3-MAPK and Hog1-MAPK in WT, ∆Mr-OPY2 and ∆Mr-STE50 using their respective antibodies. During saprophytic growth in SDY, phosphorylation levels of Hog1-MAPK and Fus3-MAPK were similar in ∆Mr-OPY2 and ∆Mr-STE50, and slightly lower than in the WT (Fig. 6). Phosphorylation of Fus3-MAPK and Hog1-MAPK was reduced in ∆Mr-OPY2 and ∆Mr-STE50 growing in hemolymph or under high osmotic stress (Fig. 6). Thus, in hemolymph, phosphorylation of Fus3-MAPK was reduced 1.7-fold and 11-fold relative to WT in ∆Mr-STE50 and ∆Mr-OPY2, respectively. During appressorial formation, phosphorylation of Hog1-MAPK was reduced in the two mutants; whereas phosphorylation of Fus3-MAPK was unchanged compared wit the WT (Fig. 6).

**Mr-OPY2 negatively regulates AFTF1 during cuticle penetration.** RNA-seq was used to compare WT and ∆Mr-OPY2 transcriptomes of undifferentiating hyphae in SDY and hemolymph cultures, and germlings differentiating appressoria on locust wings. WT and ∆Mr-OPY2 grown in SDY, hemolymph and locust cuticle differed by 82, 5 and 39 DEGs (differentially expressed genes), respectively. Only one DEG encoding a hypothetical protein was shared in the three growth conditions (Fig. 7a), indicating that Mr-OPY2 regulates distinctive subsets of genes during saprophytic growth, cuticle penetration and colonization of the hemocoel.

During differentiation of appressoria and cuticle penetration, a single transcription factor (MAA_08552) with a $Zn_2Cys_6$ fungal type DNA binding domain, designated as AFTF1, was differentially expressed between WT and ∆Mr-OPY2. qRT-PCR analysis

showed Aftf1 was upregulated ~ 31 fold (P < 0.01, Tukey's test in one-way ANOVA) on insect cuticle relative to hemolymph and SDY (Fig. 7b). On locust wings, expression of Aftf1 in ∆Mr-OPY2 was 7.2-fold higher than WT (P < 0.01, Tukey's test in one-way ANOVA), but it was reduced > 11-fold (P < 0.01, Tukey's test in one-way ANOVA) in ∆Mr-STE50 and the two MAPK mutants (∆Meor-Fus3 and ∆Mero-Slt2) (Fig. 7c). Both ∆Meor-Fus3 and ∆Mero-Slt2 are impaired in appressorial formation[24].

In order to investigate the functions of Aftf1, we deleted its ORF to produce ∆Aftf1, which was complemented with its genomic clone to produce C-∆Aftf1. Confirmation of gene disruption and mutant complementation is shown in Supplementary Fig. 2. We also constructed strains (designated as Aftf1[OE]) overexpressing Aftf1 driven by the constitutive promoter (Ptef) of the translation elongation factor gene from Aureobasidium pullulans[34]. Germlings of two randomly selected Aftf1[OE] transformants significantly (P < 0.01, Tukey's test in one-way ANOVA) upregulated (~ 300-fold) transcription of Aftf1 compared with WT when differentiating appressoria on locust cuticle (Supplementary Fig. 9a). No intra-strain differences were found between the two isolates in any assay, and so representative data for a single isolate is provided. Conidiation and saprophytic growth of Aftf1[OE] and ∆Aftf1 on PDA resembled the WT (Supplementary Fig. 9b, c). However, appressorial formation by ∆Aftf1, and to a lesser extent Aftf1[OE], was significantly delayed (P < 0.05, Tukey's test in one-way ANOVA) (Fig. 7d), significantly (P < 0.05, Tukey's test in one-way ANOVA) reducing pathogenicity compared to the WT (Fig. 7e). In all the above assays, there were no significant (P > 0.05, Tukey's test in one-way ANOVA) differences between the WT and C-∆Aftf1 (Fig. 7 and Supplementary Fig. 9). Therefore, we conclude that the precise level of AFTF1 protein controlled by Mr-OPY2 is important for appressorial formation and pathogenicity.

Appressorial production is associated with synthesis of key effector proteins influencing virulence[24]. RNA-seq analysis showed that four cuticle-degrading proteases (MAA_00460, maa_07827, MAA_07828 and MAA_09637) were downregulated in ∆Mr-OPY2 during penetration of the cuticle. In addition, a cell wall surface protein Mas1 (MAA_08589) was upregulated; its ortholog is an important factor for appressorial formation in Magnaporthe oryzae[35]. qRT-PCR analysis showed that the protease MAA_09637 was also down-regulated in ∆Aftf1, suggesting that AFTF1 could regulate a subset of the genes

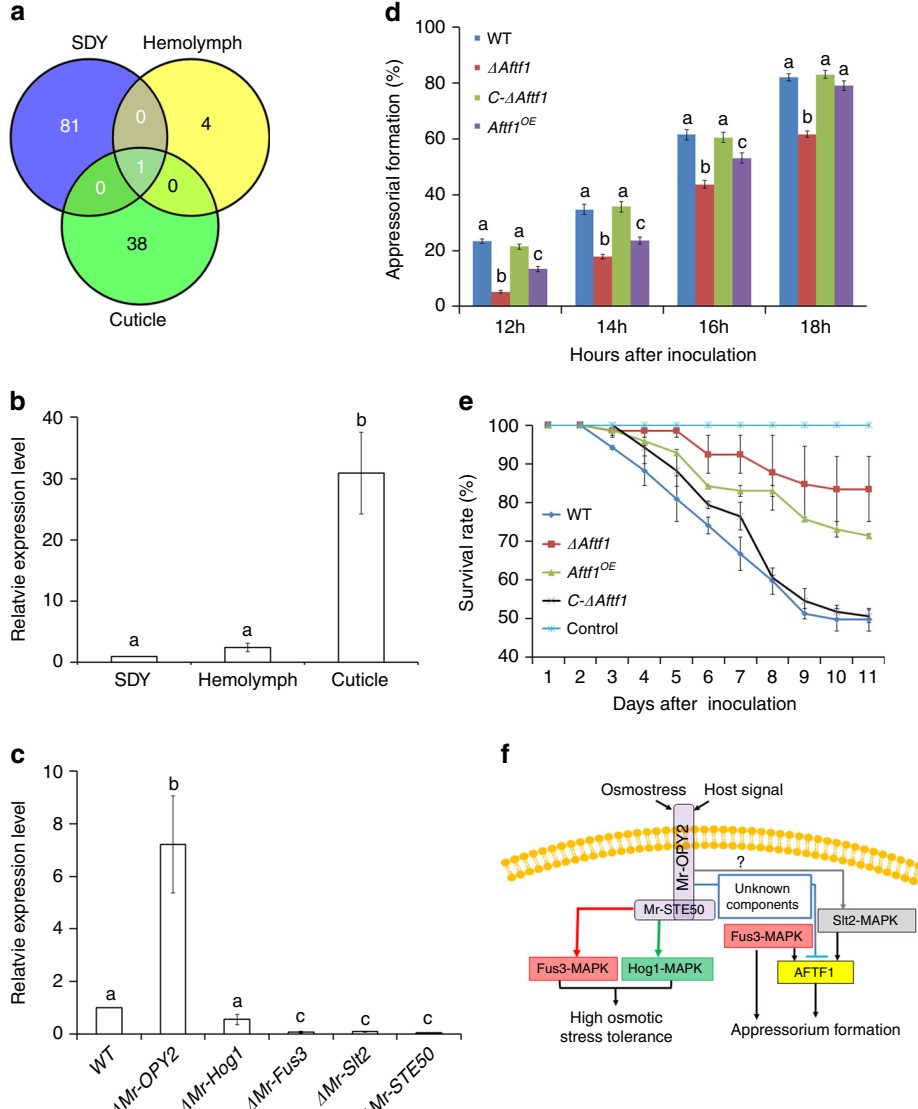

**Fig. 7** Identification and characterization of the appressorial formation transcription factor AFTF1 and its negative regulation by Mr-OPY2. **a** Venn diagram of RNA-seq analysis showing the distribution of shared DEGs in transcriptomes of WT and *ΔMr-OPY2* during growth in SDY (SDY), the insect cuticle (Cuticle) and hemolymph (Hemolymph). Two biological repeats were established for each treatment. **b** qRT-PCR analysis of *Aftf1* expression by the WT grown in hemolymph and on locust cuticle relative to expression during saprophytic growth in SDY (which is set to 1 in the figure). **c** qRT-PCR analysis of *Aftf1* expression by the WT, *ΔMr-OPY2*, *ΔMr-STE50*, *ΔMero-Hog1*, *ΔMero-Fus3* and *ΔMero-Slt2* during appressorial formation. The expression level in the WT is set to 1. The qRT-PCR analyses were repeated three times with three replicates per repeat. Data are expressed as the mean ± SE. Values with different letters are significantly different (*n* = 3, *P* < 0.05, Tukey's test in one-way ANOVA). **d** The % germlings differentiating appressoria on a hydrophobic plastic surface in the WT, *ΔAftf1* (the *Aftf1* disruption mutant), *C-ΔAftf1* (the complemented strain of *ΔAftf1*) and *Aftf1OE* (a strain overexpressing *Aftf1*). Appressorium formation assays were repeated three times with three replicates per repeat. Data are expressed as the mean ± SE. At each time point, values with different letters are significantly different (*n* = 3, *P* < 0.05, Tukey's test in one-way ANOVA). **e** Survival curves of *G. mellonella* larvae infected by WT, *ΔAftf1*, *C-ΔAftf1* and *Aftf1OE*. Data are expressed as the mean ± SE. Control: insects treated with 0.01% Triton X-100 solution. The bioassays were repeated three times with 40 insects per repeat. **f** A schematic model of Mr-OPY2-mediated regulation of appressorial formation and osmotic stress tolerance. Mr-OPY2 regulates the phosphorylation level of Fus3 and Hog1-MAPK under high osmotic stress. During appressorial formation, Fus3-MAPK regulates the expression level of *Aftf1*, but regulation of *Aftf1* by Mr-OPY2 is mediated by unidentified components. Slt2-MAPK also regulates *Aftf1*, but it remains to be determined whether Mr-OPY2 is involved in such regulation

regulated by Mr-OPY2 (Supplementary Fig. 10). The expression level of these five genes did not differ significantly (*P* > 0.05, Tukey's test in one-way ANOVA) between WT and *Atf1*OE.

## Discussion

Fungi can live as saprophytes in diverse habitats, on virtually any carbon source, or in commensal, mutualistic or parasitic relationships with plants, animals, insects or other fungi. The mechanisms that regulate transitions between saprophytic, symbiotic and pathogenic growth phases are poorly understood[36], with the exception of a transcriptional regulator, Wor1/Ryp1, that governs lifestyle transitions in human pathogenic fungi, *C. albicans* and *Histoplasma capsulatum*[37,38], and several plant pathogens including *Fusarium verticillioides* (Sge1)[36]. In this study, we report that the anchor membrane protein Mr-OPY2 is

dispensable for saprophytic growth of *M. robertsii*, but is essential for this fungus to initiate development of infection structures (appressoria), which are a prerequisite for infection. Therefore, Mr-OPY2 regulates the transition of *M. robertsii* from saprophyte to pathogen; to our knowledge, this is the first demonstration that OPY2 regulates the initiation of infection in a pathogenic fungus. Homologs of Mr-OPY2 are conserved in plant and mammal pathogens, suggesting a potential role in regulating pathogenesis in other fungi, especially the many plant pathogenic fungi (e.g. *M. oryzae*) that also differentiate appressoria.

The mechanism by which Mr-OPY2 controls *M. robertsii*'s saprophyte-to-pathogen transition involves precise regulation of its protein level, with higher levels during cuticular penetration than saprophytic growth. The *Mr-OPY2* disruption mutant, and a manipulated strain (*P404-Mr-OPY2*) with a constantly low Mr-OPY2 protein level, were both impaired in appressorial formation and thus in pathogenicity. This confirms that upregulation of the Mr-OPY2 protein level is necessary for infection processes. Conversely, a low level of Mr-OPY2 protein is important for normal saprophytic growth because forcefully elevating Mr-OPY2 (the strain *P683-Mr-OPY2*), with a constitutive promoter produced an abnormal colony phenotype with reduced conidial yield. *P683-Mr-OPY2* was also impaired in appressorial formation, indicating that constitutively expressing high levels of Mr-OPY2 protein destabilizes appressorial formation. Mr-OPY2 is also upregulated in the insect hemolymph, but its role following penetration of the cuticle remains to be elucidated because Δ*Mr-OPY2* resembled WT in its ability to colonize the hemocoel. Western blot analyses of Mr-OPY2 showed two bands in all strains (including WT) that express Mr-OPY2, which were absent in the deletion mutant of *Mr-OPY2*. One band is close to the theoretical molecular weight of Mr-OPY2 (45.5 kDa), whereas the other is ~50 kDa. The heavier band could result from post-translational modifications; the difference in the intensity of the two bands should result from the difference between the amount of protein modified and unmodified. *C. albicans* and *S. cerevisiae* OPY2 proteins were similarly bigger than their theoretical molecular weights due to glycosylation[39,40].

The precise regulation of Mr-OPY2 protein levels is achieved via alternative transcription start sites. During pathogenesis, two *Mr-OPY2* transcripts were produced; the abundant long transcript was inefficiently translated because its 5′UTR has two small uORFs that were proven by two independent experiments to reduce translation efficiency of the downstream major ORF. As described in other studies[29–31], small uORFs can modulate ribosomal access to the AUG start codon of the major ORF, decreasing its translation efficiency. Although the short transcript of *Mr-OPY2* has a lower copy number than the long one, it is more efficiently translated. Thus, although the long transcript was abundant during both saprophytic growth and pathogenesis, increased levels of the Mr-OPY2 protein during pathogenesis derived from the short transcript. We are not aware of any other reports on regulation of OPY2 protein levels via alternative transcription start sites. However, alternating transcription start sites is a common mechanism for regulating gene expression in fungi[41].

Only a single novel transcription factor (AFTF1) is regulated (negatively) by Mr-OPY2 during cuticle penetration. Homologs of AFTF1 are also found in other fungi such as *Fusarium oxysporum* (EWY87630) and *M. oryze* (XP_003715433). While upregulation of AFTF1 is critical for appressorial formation, the process was delayed by overexpressing AFTF1. Therefore, a precisely modulated optimal level of AFTF1 is needed to choreograph appressorial formation with Fus3- and Slt2-MAPK upregulating *Aftf1* during cuticle penetration and Mr-OPY2 ensuring that the level of AFTF1 is not too high for optimum

effect. The precise regulation of Mr-OPY2 is presumably an adaptation to facilitate the precise regulation of AFTF1. The phosphorylation level of Fus3-MAPK is not controlled by Mr-OPY2, suggesting that Mr-OPY2 does not regulate AFTF1 via Fus3-MAPK. Yeast two-hybrid assays showed that the MAPKKK (Bck1) of the Slt2-MAPK cascade interacts with Mr-STE50, which in turn interacts with Mr-OPY2, suggesting that Mr-OPY2 could regulate AFTF1 via Slt2-MAPK. Mr-OPY2 regulates phosphorylation of Hog1-MAPK during appressorial formation, but Hog1-MAPK does not regulate expression of *Aftf1* indicating that Mr-OPY2 has additional functions besides regulating AFTF1 (Fig. 7f).

In addition to regulating pathogenesis, OPY2 is also necessary for *M. robertsii* to tolerate high osmotic stress, but as this stress did not alter Mr-OPY2 protein levels, low levels may be sufficient for stress resistance. Mr-OPY2 regulates *M. robertsii*'s tolerance to high osmotic stress via the Hog1-MAPK and Fus3-MAPK pathways (Fig. 7f). *S. cerevisiae* Opy2 protein also mediates tolerance to high osmotic stress via Hog1-MAPK, but the *Opy2* deletion mutant can grow under the osmotic stress because the redundant SSK branch also regulates Hog1-MAPK to tolerate osmotic stress[26,40,42]. The Opy2 protein of *C. albicans* is not involved in tolerance to high osmotic stress[40], or *C. albicans*'s pathogenicity against a natural host, but by an unknown mechanism does facilitate its ability to kill a non-natural host (the insect *G. mellonella*) following injection[40]. In contrast, we found that OPY2 is dispensable for hemocoel infection by the specialized insect pathogen *M. robertsii*. Therefore, the functions of OPY2 proteins have clearly diversified in fungi, along with their lifestyles. Our phylogenetic analysis confirmed that the OPY2 proteins from unicellular Ascomycota yeasts (including *S. cerevisiae* and *C. albicans*) are phylogenetically distant from their homologs in filamentous Ascomycota and Basidiomycota fungi.

In conclusion, we report that Mr-OPY2 controls the saprophyte-to-pathogen transition of the insect pathogen *M. robertsii* by negatively regulating the transcription factor AFTF1. This work describes a new circuit regulating fungal infection, and advances our insight into the mechanisms underpinning fungal lifestyle transitions.

## Methods

**Gene cloning and disruption**. The flanking sequences of T-DNA were cloned by YADE (Y-shaped adaptor dependent extension) from *M. robertsii* mutants generated by T-DNA insertion as previously described[22]. *M. robertsii* ARSEF2575 and *M. acridum* ARSEF324 were obtained from the Agricultural Research Service Collection of Entomopathogenic Fungi. The primers used in this study are listed in Supplementary Table 2.

Genes were disrupted based on homologous recombination using our previously developed high-throughput gene disruption methodology[43]. Disruption plasmids were constructed using Ppk2-OSCAR-GFP and pA-Bar, and confirmation of gene disruption was performed as described[43]. To complement a gene disruption mutant, the genomic clone of the gene containing the promoter region, ORF and termination region was amplified by PCR and inserted into the pPK2-Sur-GFP[43]. High fidelity Taq DNA polymerase (KOD Plus Neo, Osaka Japan) was used in PCR reactions to clone DNA fragments, which were confirmed by DNA sequencing. *Agrobacterium tumefaciens* AGL1 was used for fungal transformation as previously described[44].

**Histoimmunochemical staining and western blot analysis**. In order to prepare an antiserum to Mr-OPY2, a predicted antigenic region (PGPNASPDQIKKHRD), residues 59–73 of the *N*-terminal domain, was synthesized and conjugated with keyhole limpet hemocyanin. The antibodies were raised in New Zealand White rabbits (Sigma, USA). Total cell membrane protein from fungal cells was prepared using the Mem-PER Eukaryotic Membrane Protein Extraction Reagent Kit (Thermo Scientific, USA), followed by purification with the Pierce sodium dodecyl sulfate polyacrylamide gel electrophoresis (SDS-PAGE) Sample Prep Kit (Thermo Scientific). Two identical SDS-PAGE gels were set up, one of which was used for western blot analysis, and the other one was stained with Coomassie Brilliant Blue in order to visualize the sample loading. The antiserum (200 mg ml$^{-1}$) was used at a 1:500 dilution to detect Mr-OPY2 protein with western blot analysis.

To analyze phosphorylation levels of Fus3 and Hog1-MAPK in *M. robertsii*, cell proteins were prepared with TCA (trichloroacetic acid) buffer as described[45], and blotted to polyvinylidene difluoride membranes (Bio-Rad, USA). Rabbit anti-phospho-p38 MAPK (Thr180/Tyr182) (Catalogue number: 9211), anti-phospho-p44 MAPK (Catalogue number: 9101) and anti-p38 MAPK (Catalogue number: 9212) (Cell Signaling Technology, MA, USA) were used at a 1:1000 dilution to detect phosphorylation of Hog1-MAPK, Fus3-MAPK and the protein level of Hog1-MAPK, respectively. To prepare an antiserum to the Fus3-MAPK protein, a predicted antigenic region (CHDPEDEPTAPPTP) was synthesized, and the antiserum was raised in New Zealand White rabbit (HUABIO, Hangzhou, China). The antiserum (0.4 mg ml$^{-1}$ IgG) was used at a 1:1000 dilution to detect the Fus3-MAPK protein. β-tubulin was used as a loading control, which was detected using its mouse antibody (Catalogue number: M1305-2, HUABIO) at a 1:2000 dilution. Bound primary antibodies were revealed using horseradish peroxidase-conjugated goat anti-rabbit IgG (Catalogue number: HA1006, HUABIO) or goat anti-mouse IgG (Catalogue number: HA1006, HUABIO) with a 1:4000 dilution, and a chemiluminescence detection system (Clarity Western ECL, Bio-Rad).

Appressoria on the hydrophobic surface of a diagnostic slide (Live Focus, Jiangsu, China) were prepared for IIF as described[46]. Mr-OPY2 antiserum (200 mg ml$^{-1}$) was diluted 200-fold for primary labeling, and FITC-conjugated-goat anti-rabbit IgG (Catalogue number: HA1004, HUABIO) was used for secondary labeling.

All Histoimmunochemical staining and Western blot analyses were repeated at least three times, and representative results are shown.

**Assays of pathogenicity and conidiation and stress tolerance**. Bioassays were conducted using last instar *G. mellonella* larvae (Ruiqingbait Co., Shanghai, China) by topical application or direct injections as described (Zhao et al., 2014). Tolerance to high osmotic stress, oxidative stress and cell wall-disturbing stress was assayed as previously described[24]. Determination of conidial yields and observation of conidiophores were conducted as previously described[24]. All assays were repeated three times.

**Site mutagenesis**. PCR mutagenesis was used to mutate the three AUGs into GUGs and thereby delete the uORFs in the 5′UTR of *Mr-OPY2-L*. The *Mr-OPY2* genomic clone (*gMr-OPY2*) containing its exons, introns and terminator region was cloned by PCR using high fidelity Taq DNA polymerase into the pUCm-T (Takara, Dalian, China) to produce the plasmid PUC-gMr-OPY2. Two complementary primers (Supplementary Table 2) containing mutated sites flanked by 20 bases on each side were designed and used to amplify PUC-gMr-OPY2. The PCR product was then digested with DpnI (NEB, England) to cut the parental plasmid DNA, whereas the nascent PCR DNA was left intact. The PCR products were then incorporated into the *E. coli* strain that has an intact dam methylase system. The resulting plasmids thus contained no uORFs, and this was confirmed by DNA sequencing. The AUG at −167 bp (167 bp upstream of the AUG of the major ORF) in *gMr-OPY2* was initially mutated to produce *gMr-OPY2*$^{ΔAUG167}$, and the two linked AUGs at −363 bp and −366 bp in *gMr-OPY2*$^{ΔAUG167}$ were then mutated to form *gMr-OPY2*$^{ΔAUGs}$.

**Constructing strains with the Mr-OPY2 gene driven by Pgpd**. To analyze the translation efficiency of the transcripts *Mr-OPY2-L*, *Mr-OPY2-S* and *Mr-OPY2-OPY2-L*$^{ΔAUGs}$, each was placed downstream of the promoter *Pgpd-NUTR* (the *gpd* gene's 5′UTR excluded) of the *gpd* gene from *M. acridum* ARSEF324[33]. To do this, *Pgpd-NUTR* was amplified using the genomic DNA of *M. acridum* ARSEF324 as the template, which was then inserted into the *NdeI/Bam*HI sites in the plasmid pBARGPE1 (obtained from Fungal Genetic Stock Center) to replace the *Aspergillus nidulans gpd* promoter, resulting in the plasmid pPgpd-NUTR. The DNA fragment containing *Pgpd-NUTR* and the *TtrpC* terminator was excised from pPgpd-NUTR with *Spe*I/*Sma*I and inserted into *Spe*I/*Eco*RV sites of the plasmid pPK2-Sur-GFP[43] to produce the plasmid pPK2-Pgpd-NUTR. The transcripts of *Mr-OPY2-L*, *Mr-OPY2-S* and *Mr-OPY2-OPY2-L*$^{ΔAUGs}$ were cloned by PCR, digested with *Eco*RI/*Sma*I and inserted into *Eco*RI/*Eco*RV sites downstream of the promoter *Pgpd-NUTR* in pPK2-Pgpd-NUTR to produce plasmids pPK2-Mr-OPY2-L, pPK2-Mr-OPY2-S and pPK2-Mr-OPY2-OPY2-L$^{ΔAUGs}$, respectively. The three plasmids were then transferred into *Agrobacterium tumefaciens* and transformed into the *Mr-OPY2* deletion mutant Δ*Mr-OPY2* to produce the strains Δ*Mr-OPY2*::*Mr-OPY2-S*, Δ*Mr-OPY2*::*Mr-OPY2-L* and Δ*Mr-OPY2*:: *Mr-OPY2-L*$^{ΔAUGs}$.

The strains *P404-Mr-OPY2* and *P683-Mr-OPY2* were constructed by placing the major ORF downstream of either the 404 bp promoter (P404) or the 683 bp promoter (P683) of the *M. acridum gpd* gene (including the 5′UTR of the *gpd* gene)[33]. The P404 and P683 promoters were cloned by PCR using the genomic DNA of *M. acridum* ARSEF324 as the template, and digested with enzymes *NdeI/Bam*HI to replace the *gpd* promoter of *A. nidulans* in the plasmid pBARGPE1 and produce plasmids pP404 and pP683. The DNA fragment containing the promoter (P404 or P683) and trpC terminator region was excised with *Spe*I/*Sma*I and inserted into the *Spe*I/*Eco*RV sites in the pPK2-Sur-GFP plasmid to produce pPK2-P404 and pPK-P683, respectively. The mORF of Mr-OPY2 was cloned by PCR and excised with *Eco*RI/*Kpn*I, which was then inserted into the corresponding sites downstream of the promoters P404 and P683 in pPK2-P404 and pPK-P683 to form the plasmids pPK2-P404-Mr-OPY2 and pPK2-P683-Mr-OPY2, respectively.

pPK2-P404-Mr-OPY2 and pPK2-P683-Mr-OPY2 were then transferred into *A. tumefaciens* and transformed into the *Mr-OPY2* deletion mutant Δ*Mr-OPY2* to produce the strains *P404-Mr-OPY2* and *P683-Mr-OPY2*.

**Transcriptomic analysis using RNA-Seq**. RNA-seq was used to profile transcriptomes during saprophytic growth in SDY, appressorial formation (30 h post inoculation of conidia onto the hindwings of *L. migratoria*) and growth in *B. mori* hemolymph. Total RNA was extracted with TRIzol reagent (Life Technologies, USA). For saprophytic growth, conidia (1 × 10$^6$ conidia/ml) were grown at 26 °C for 36 h in SDY. For appressorial formation, hindwings of *L. migrattoria manilensis* were surface sterilized in 1% bleach as previously described[21]. A conidial suspension (200 μl of 1 × 10$^5$ conidia/ml) was applied to each hindwing which was then placed on a 1% water agar plate, and incubated at 26 °C for 30 h. For growth in the hemolymph, mycelium from a 36 h SDY culture was collected by filtration and washed with sterile water three times. One gram of mycelium was inoculated into 12 ml of *B. mori* hemolymph and incubated for 12 h at 26 °C with 100 rpm shaking. The mycelium was then collected by filtration for RNA extraction. Locust wings and silkworm hemolymph were prepared as described[24]. Two biological repeats were established for each treatment.

Construction of libraries and sequencing with the Illumina HiSeq 2500 platform were performed by Novogene (Beijing, China). After paired-end sequencing, clean reads were obtained using the NGS QC Toolkit and mapped to the draft genome sequence of *M. robertsii* ARSEF 23[16] using the programs TopHat 2.0.6[47]. Reads that aligned uniquely to the reference sequence were used for gene expression quantification using the RPKM method[48]. Differential expression analysis was performed with DESeq software (version 1.18.0)[49], and an adjusted *P*-value 0.05 (Benjamini–Hochberg method).

**Construction of the strain overexpressing *Aftf1***. The strain overexpressing *Aftf1* was constructed by placing its ORF downstream of the constitutive promoter *Ptef* of the translation elongation factor gene from *A. pullulans*[34]. To do this, the promoter *Ptef* was cloned by PCR using the plasmid pPK2-Sur-GFP[43] as the template, and digested with enzymes *NdeI/Bam*HI to replace the *gpd* promoter of *A. nidulans* in the plasmid pBARGPE1 and produce the plasmid pPtef. The DNA fragment containing the promoter and trpC terminator region was excised with *Spe*I/*Sma*I and inserted into the *Spe*I/*Eco*RV sites in the pPK2-Sur-GFP plasmid to produce pPK2-Ptef. The ORF of *Aftf1* was cloned by PCR and excised with *Eco*RI/*Eco*RV, which was then inserted into the corresponding sites of pPK2-Ptef to produce pPK2-Ptef-Aftf1. pPK2-Ptef-Aftf1 was then transferred into *A. tumefaciens* and transformed into the WT *M. robertsii*. Overexpression of *Aftf1* was confirmed by qRT-PCR as described below.

**Gene cloning and gene expression analysis**. RLM RT-PCR kit was used to clone intact mRNA with a 7-methyl guanosine cap structure (Roche).

For quantitative RT-PCR, complementary DNAs (cDNAs) were synthesized with total RNAs (800 ng per reaction) with ReverTra AceqPCR RT Master Mix (Toyobo, Japan). Each cDNA was diluted 20-fold, and used as a template for qRT-PCR analysis using Thunderbird SYBR qPCR Mix without ROX (Toyobo). The genes *act* and *tef* were used as internal standards[50]. The relative expression level of each gene was determined using the $2^{-ΔΔCt}$ method[51].

Northern blot analyses were conducted using standard methods[52]. The ORF of the *Mr-OPY2* was used as the probe, which was labelled with DIG-DNA Labelling Kit (Roche). Northern blot and qRT-PCR analyses were repeated three times.

**Yeast two-hybrid and autoactivation assays**. Yeast competent cells were prepared using the Yeast-maker Yeast Transformation System 2 Kit (Clontech, USA), and yeast two-hybrid assays were conducted according to the manufacturer's instructions (Clontech). The ORFs of the tested genes were cloned by PCR and inserted into the plasmid pGADT7 (*Mr-OPY2*, *Mero-BCK1*, *Mero-STE11* and *Mero-SSK2*) or pGBKT7 (*Mr-STE50*). The ORFs in the plasmids were confirmed by sequencing. The plasmid pGBKT7-Mr-STE50 was transformed into Y2HGold cells, and other plasmids (from pGADT7) were transformed into Y187 cells. After mating, the resulting strains were grown on the medium (SD-His-Ade-Leu-Trp) with X-α-gal and AbA (Aureobasidin A) (Takara). The autoactivation of Mr-STE50 in pGBKT7 was tested using Y2HGold cells. Yeast two-hybrid and autoactivation assays were repeated three times.

**Statistical analysis**. Tukey's honestly significant difference test in OriginPro 8.5 program was used in this study (OriginLab, USA)

**Data availability**. The nucleotide sequences of the long and short *Mr-OPY2* transcripts were deposited in the Genbank nucleotide database with accession codes KY548479 and KY548480. The RNA-seq data were deposited in the Genbank database with accession codes SRP105041. All other relevant data supporting the findings of the study are available in this article and its Supplementary Information files, or from the corresponding author upon request.

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

## Acknowledgements

This work was funded by the National Natural Science Foundation of China (31272097 and 31672078).

## Author contributions

N.G. and Y.Q. designed and performed experiments. X.C. constructed the deletion mutant of *Aftf1*. Q.Z. performed the phylogenetic analysis. G.Z., X.Z. and W.M. performed complementation of the deletion mutants and bioassays. C.X. analysed the RNA-seq data. R.J.S.L. wrote the manuscript. W.F. conceived the idea, designed and coordinated the study, and wrote the paper.

## Additional information

**Competing interests:** The authors declare no competing financial interests.

