## [Peer Review File · Nature Communications]

Reviewers' comments:

Reviewer #1 (Remarks to the Author):

Understanding fungal pathogens that have complex lifestyles in different hosts is an interesting and relevant question. Here the authors explore a signaling protein identified in an insect pathogenic fungus. They found two surprising things about the signaling protein that were not known: 1) its translation is regulated by an upstream element, which impacts its function and 2) the signaling protein interacts with a newly identified transcription factor to uniquely modulate its activity. There are many problems and errors with this paper: namely clarity in terms of helping the reader to understand the findings and put them into context. Below follow some suggestions:

In the abstract 'follow on analysis' is not clear

There are numerous typos, particularly in the figure legends.

P4, 'OPY2p' should be 'Opy2p'

Figure 1 is not particularly clear or helpful. Perhaps readers might not be surprised that Opy2 homologs exist in distantly related fungal species. Move to the supplement?

A supplemental figure showing the alignment with highlighted regions of the two Opy2s might be worth looking at.

Is Opy2's RA domain that binds to Ste50 is conserved?

Figure 2B is not that clear. Maybe it should be bigger? Hard to see the hyphal swelling. Can the CalW staining intensity be quantitated? Can the number of septum be counted?

What do the uninfected caterpillars look like?

The cross sections in panel 2C are not clear. Figure 2C is not mentioned in the text until much later in the paper.

Figure 2E is not clear. I can see fluorescence in one of the merged images but not the unmerged image (for WT hydrophobic).

Figure 3D, control for protein loading should be put next to the figure (or a subset of the bands from the gel). Same problem for 4A, Figure 4C and all other blots where this is done.

Figure 4B, given that the secondary structures are not relevant, they should be moved to the supplement.

Figure 4A, there is a background level by immunoblot in the Δ Mr-OPY2 strain. Was this subtracted for the normalization of the other samples?

Figure 4A; it looks like the lower band is of higher intensity than the upper band. Why would that be expected to happen?

Were the transcript levels checked for the data provided in Fig. 5A?

Cells lacking Opy2 are not virulent (Figure 1). Cells expressing more Opy2 are less virulent (Figure 5). This is confusing and not clearly explained or discussed in this section by the authors. What are different assays used in Figs. 1 and 5?

The rationale for the experiments in several of the results sections is not well introduced. Why look at salt sensitivity? This needs to be introduced better.

In yeast, *opy2* mutants are not sensitive to osmotic stress because of a redundant branch of the HOG pathway (SSK branch). Is there an SSK branch in this fungus?

A control is missing from the two hybrid: only containing one (bait) or the other (prey) plasmid to show that auto-activation is not occurring.

Yeast *Opy2* does not directly interact with MAPK cascade components.

I'm not sure their model accurately captures the data, especially see Figure 7.

Reviewer #2 (Remarks to the Author):

Alternative transcription start site selection of the Mr-OPY2 gene controls the transition from saprophyte to pathogen in *Metarhizium robertsii*

The paper presents evidence that alternative transcription may play a role in regulation of the membrane protein-encoding gene Mr-OPY2. The Mr-OPY2 protein is produced abundantly and is required for appressorium formation in the insect pathogenic fungus *Metarhizium*, which is essential for insect infection. By contrast, low level expression of the gene is associated with saprophytic growth of the fungus. The authors suggest that during saprophytic growth the fungus produces a single long transcript that contains 5'UTR microORFs that suppress translation of the larger downstream major ORF. On insect cuticle, a shorter transcript leads to production of Mr-OPY2 protein. This appears to act as a negative regulator of the AFTF1 transcription factor involved in regulating appressorium development.

The paper is interesting and is a reasonably comprehensive study of the potential means by which the Mr-OPY2 is regulated. However, what is less clear to me is whether this gene is actually a pivotal regulator of insect infection, or whether this form of regulation is more widespread within a broader range of genes associated with insect infection. This leaves this reviewer wondering whether the key finding is the mode of regulation (which is actually quite well known in filamentous fungi) in the context of insect infection, or if it is the actual regulation of Mr-OPY2 that is the major finding. Mr-OPY2 serves a similar function in *Metarhizium* to the counterpart in yeast apart from its means of regulation and the putative interaction with AFTF1. Given that I find it hard to distinguish between the two possibilities, coupled with the fact that there are many technical issues with the paper, I am not really convinced that it warrants publication in *Nature Communications*.

Major concerns

1. Figure 1. It is not clear what the key (0.7) relates to precisely. The figure legends suggests this relates to the estimated number of amino acid substitutions, but how does this actually relate to the tree. Similarly, the figures in the Table within Figure 1B are not clear to me (statistics, but which?).
2. Figure 2B. This is too small to really show differences in appressorium morphology and they are not at all clear from the Figure. No scale bar. Figure 2C also needs a scale bar in both panels. The differences in mycelial colonization in the lower panels are not at all clear and no indication of replication or quantification of this difference is provided.
3. Figure 3. A No size marker or context at all. No indication of replication of experiment or quantification. Figure 3D No size marker or indication of replication. Figure 3E completely inadequate to show any differences. None of the differences referred to in the text are visible in

this figure. No scale bar. Too small to be of any use, not consistent with text description.

4. Figure 4A. No size marker in lower panel. No context, no quantification, no indication of replication of the experiment. Figure 4C. No size marker. No loading control. No evidence of replication.

5. Figure 4D is mentioned in the text. Page 7, line 151 stating that Mr-OPY2 is 1.4 fold higher in hemolymph than in SDY. This panel is not present in the Figure.

6. Figure 5A. No loading control or size marker (not much use giving this in the supplement). No evidence of reproducibility. Figure 6B. No scale bar. Morphological differences stated in text not clear. Statistical analysis of conidiation differences not provided.

7. Figure 6A needs scale bars. Not really clear at all. The micrograph are no clear and too small. Figure 6C has no loading control or size markers. Figure 6D is not at all clear as a Y2H analysis. Where is dilution and high stringency versus medium/low stringency result. Inadequate control.

8. Figure 7. The differences in phosphorylation are key to the conclusions of the paper and yet this figure is really very poor. No indication of reproducibility is provided. No quantification of differences is provided apart from the 'ratios' which are not explained. This needs to be analyzed properly with replicates, given the critical nature of the data presented.

9. Supplemental Figures. I find the 'YES' 'NO' panels for each of the gels presented to be confusing/inadequate to illustrate the cloning strategies. No size markers on any of the gels shown also compounds the lack of clarity. No size markers provided on any of the protein gels.

For this reviewers, the lack of rigor in the experimental results presented are a real problem in interpreting the data. I cannot really decide whether the finding is novel and a 'new insight into fungal lifestyle transitions' given the poor quality of the data presented. I think they need to evidence the reproducibility of their findings and also provide evidence that Mr-OPY2 is the key gene in this context that they claim.

Point-to-point response to reviewer's comments

The reviewer's comments are in plain text and our responses are in **bold**.

Reviewer #1 (Remarks to the Author):

Understanding fungal pathogens that have complex lifestyles in different hosts is an interesting and relevant question. Here the authors explore a signaling protein identified in an insect pathogenic fungus. They found two surprising things about the signaling protein that were not known: 1) its translation is regulated by an upstream element, which impacts its function and 2) the signaling protein interacts with a newly identified transcription factor to uniquely modulate its activity. There are many problems and errors with this paper: namely clarity in terms of helping the reader to understand the findings and put them into context.

Response: To improve the clarity, we have made substantial changes to the text, figures and their legends.

Below follow some suggestions:

1. In the abstract 'follow on analysis' is not clear

Response: "follow-on analysis" mainly involved identification of the transcription factor AFTF1 and functional characterization of AFTF1 using gene disruption and qRT-PCR. We clarified this in the revised manuscript.

2. There are numerous typos, particularly in the figure legends.

Response: corrected.

3. P4, 'OPY2p' should be 'Opy2p'

Response: we used *OPY2* for the protein's name, and *OPY2* (italicized) for the gene's name. This is common in fungi as compared to say bacteria where the gene is usually lower case. As described in the link below the general rule that gene symbols are italicized and protein symbols are not italicized holds true regardless of the type of organism, but there are several variations among organisms in the composition and capitalization of alphanumeric characters within the gene and protein symbols.

<http://www.biosciencewriters.com/Guidelines-for-Formatting-Gene-and-Protein-Names.aspx>

4. Figure 1 is not particularly clear or helpful. Perhaps readers might not be surprised that Opy2 homologs exist in distantly related fungal species. Move to the supplement?

Response: The figure is moved to supplemental. However, in addition to indicating that OPY2 homologs exist in distantly related fungal species, this figure also shows that the OPY2s of Ascomycota yeast's cluster in a clade that is separate from those of filamentous Ascomycota and Basidiomycota fungi.

5. Is Opy2's RA domain that binds to Ste50 is conserved?

Response: Yes, we found CR-A and CR-D domains in Mr-OPY2, and this information is provided in the revised manuscript.

6. Figure 2B is not that clear. Maybe it should be bigger? Hard to see the hyphal swelling. Can the CalW staining intensity be quantitated? Can the number of septum be counted?

Response: the definition of the original figure was reduced by converting it to a pdf for initial review. Nevertheless, we have prepared a new figure (figure 1 in the revised manuscript) that is considerably bigger than the original. It is not possible to accurately quantify CalW staining intensity. The septa can be counted, and new data is added.

7. What do the uninfected caterpillars look like?

Response: The infected insects are mummified and can be sectioned. The uninfected insects have liquid hemolymph in their body cavity; they can be sectioned but their bodies collapse in on themselves so don't make good pictures.

The cross sections in panel 2C are not clear. Figure 2C is not mentioned in the text until much later in the paper.

Response: We prepared new high quality figures (Figure 1 in the revised manuscript) and labeled parts of the cross section for clarity. In order for readers to understand the data more easily, we grouped the bioassay data into figure 1. The components of figure 1 are now mentioned in sequence.

8. Figure 2E is not clear. I can see fluorescence in one of the merged images but not the unmerged image (for WT hydrophobic).

Response: I assume the reviewer means Figure 3E in the original version. We prepared a new figure (now figure 2) that is bigger than the original one. However, a point of this figure is that fluorescence will be weak when the WT is grown in 1/2SDY because the level of Mr-OPY2 protein is low as in 2E. In the nature of these experiments it is harder to see fluorescence in WT when it is grown in 1/2 SDY.

In the original picture (Fig.3E), the length of germlings grown in 1/2SDY was shorter than the appressorium-forming germling. We redid the Histoimmunochemical staining experiments with hyphae grown for longer in 1/2SDY so that their length is similar to appressorium-forming germlings. We found no difference in the fluorescent intensity of long hyphae (in the new images) and short hyphae (in the images in the first submission).

9. Figure 3D, control for protein loading should be put next to the figure (or a subset of the bands from the gel). Same problem for 4A, Figure 4C and all other blots where this is done.

Response: We moved all loading control gels (a subset of bands including the target protein) to be aligned in figs with their respective blots.

10. Figure 4B, given that the secondary structures are not relevant, they should be moved to the supplement.

Response: Agreed-moved to the supplement.

11. Figure 4A, there is a background level by immunoblot in the Δ Mr-OPY2 strain. Was this subtracted for the normalization of the other samples?

Response: No, the Δ Mr-OPY2 strain was used as a negative control. WT was set to 1, and other samples except the Δ Mr-OPY2 strain were relative to WT. This is mentioned in the revised legends.

12. Figure 4A; it looks like the lower band is of higher intensity than the upper band. Why would that be expected to happen?

Response: Yeast OPY2 proteins can be glycosylated (de Dios et al., 2013, Fungal Genetics and Biology). The upper Mr-OPY2 protein is thus likely modified by glycosylation, and the difference in band intensity could result from the difference between the amounts of protein modified and unmodified.

This is now discussed in the discussion section.

13. Were the transcript levels checked for the data provided in Fig. 5A?

Response: The transcript levels were checked with qRT-PCR and results are now provided in the revised manuscript.

14. Cells lacking Opy2 are not virulent (Figure 1). Cells expressing more Opy2 are less virulent (Figure 5). This is confusing and not clearly explained or discussed in this section by the authors. What are different assays used in Figs. 1 and 5?

Response: Additional discussion has been added to clarify this point.

Fig.1 (now figure S1) in the original manuscript was about phylogenetic analysis of fungal OPY2 proteins, and Fig. 5 (now figure 4) showed precise regulation of Mr-OPY2 protein levels is important for saprophytic growth and infection.

15. The rationale for the experiments in several of the results sections is not well introduced. Why look at salt sensitivity? This needs to be introduced better.

Response: The *S. cerevisiae* yeast OPY2 protein is involved in tolerance to high osmotic stress (e.g. Yamamoto et al., *Molecular and Cellular Biology*, 2016, 36:475-487, Wu et al., *Genes & Development* 2006, 20:734-746). We assayed the involvement of Mr-OPY2 in stress tolerance to look for commonalities with the yeast protein. Additional context is added.

16. In yeast, *opy2* mutants are not sensitive to osmotic stress because of a redundant branch of the HOG pathway (SSK branch). Is there an SSK branch in this fungus?

Response: In a previous study we demonstrated that there is an SSK pathway in *M. robertsi* comprising the MAPKKK in the Hog1-MAPK cascade (SSK2, Pbs2 and Hog1) (Chen et al. *Environ Microbiol.* 2016, 18:1048-1062). In this study, we found that STE50 directly interacts with SSK2, and the Mr-OPY2 mutant is sensitive to high osmotic stress, indicating that Mr-OPY2 regulates the phosphorylation level of Hog1-MAPK through SSK2. This is different from the functions of Opy2 in *S. cerevisiae* and *C. albicans* (Wu et al., *Genes & Development* 2006, 20:734-746; de Dios, et al., *Fungal Genetics and Biology*, 2013, 50:21-32); and Opy2 proteins in the two yeasts also have different functions in regulating the Hog1-MAPK cascade (de Dios, et al., *Fungal*

Genetics and Biology, 2013, 50:21-32). Additional discussion has been added to the revised manuscript.

17. A control is missing from the two hybrid: only containing one (bait) or the other (prey) plasmid to show that auto-activation is not occurring. Yeast Opy2 does not directly interact with MAPK cascade components.

Response: Experiments were conducted to test the auto-activation of Mr-STE50 cloned in the plasmid pGBKT7-53, and new data are provided in the revised figure (Figure 5). The reviewer is correct that the *S. cerevisiae* yeast Opy2p does not directly interact with MAPKKK; it directly interacts with STE50, which in turn interacts with MAPKKK. That is why we tested interactions between Mr-STE50 and Mr-OPY2, and between Mr-STE50 and MAPKKK.

18. I'm not sure their model accurately captures the data, especially see Figure 7.

Response: This model outlines Mr-OPY2-mediated regulation of appressorial formation and tolerance to high osmotic stress. Figure 7 (now Figure 6) shows that Mr-OPY2 controls Fus3 and Hog1-MAPK's phosphorylation level under high osmotic stress, as supported by our data. During appressorial formation, the Fus3-MAPK and Slit2-MAPK regulate AFTF1. However, Mr-OPY2 does not control phosphorylation of Fus3-MAPK. Y2H assays showed that the MAPKKK (Bck1) of the Slit2-MAPK cascade interacts with STE50, which in turn interacts with Mr-OPY2. So, Mr-OPY2 possibly regulates Slit2-MAPK. In addition, there could be other unknown components involved in regulating AFTF1.

Overall, this model is based on results from several experiments.

Reviewer #2 (Remarks to the Author):

Alternative transcription start site selection of the Mr-OPY2 gene controls the transition from saprophyte to pathogen in *Metarhizium robertsii*

The paper presents evidence that alternative transcription may play a role in regulation of the membrane protein-encoding gene Mr-OPY2. The Mr-OPY2 protein is produced abundantly and is required for appressorium formation in the insect pathogenic fungus *Metarhizium*, which is essential for insect infection. By contrast, low level expression of the gene is associated with saprophytic growth of the fungus.

The authors suggest that during saprophytic growth the fungus produces a single long transcript that contains 5'UTR microORFs that suppress translation of the larger downstream major ORF. On insect cuticle, a shorter transcript leads to production of Mr-OPY2 protein. This appears to act as a negative regulator of the AFTF1 transcription factor involved in regulating appressorium development. The paper is interesting and is a reasonably comprehensive study of the potential means by which the Mr-OPY2 is regulated.

However, what is less clear to me is whether this gene is actually a pivotal regulator of insect infection, or whether this form of regulation is more widespread within a broader range of genes associated with insect infection. This leaves this reviewer wondering whether the key finding is the mode of regulation (which is actually quite well known in filamentous fungi) in the context of insect infection, or if it is the actual regulation of Mr-OPY2 that is the major finding. Mr-OPY2 serves a similar function in *Metarhizium* to the counterpart in yeast apart from its means of regulation and the putative interaction with AFTF1. Given that I find it hard to distinguish between the two possibilities, coupled with the fact that there are many technical issues with the paper, I am not really convinced that it warrants publication in *Nature Communications*.

Response: Our paper demonstrates for the first time that Mr-OPY2 is essential for insect infection (shown in the first two result sections), that precise modulation of its protein level determines *M. robertsii*'s saprophyte-to-pathogen transition (the fifth result section), and that accurate regulation of the level of Mr-OPY2 protein is achieved at transcription (via alternative transcription start sites shown in the third result section) and translation (upstream ORFs inhibit translation efficiency of the downstream major ORF shown in the fourth result section). Furthermore, we show that Mr-OPY2 regulates phosphorylation level of Hog1- and Fus3-MAPK, and we describe a new circuit regulating fungal infectivity that contains Mr-OPY2 and AFTF1. Therefore, our major findings are the discovery of Mr-OPY2 as a key regulator of pathogenesis, the mechanism for regulating Mr-OPY2 protein levels and its implications in controlling saprophytic-to-pathogenic transition. These findings dovetail with one another, which is why they are integrated in a single paper, and we see no reason to rank their importance. We think the reason for the reviewer's uncertainty about the novelty of this paper is that they judged our findings separately in different contexts, based on which the reviewer proposed their two "possibilities".

For the first “possibility”, the reviewer thought that if the key finding is the regulation of translation by uORFs, then the paper is not novel because this regulatory mechanism is “quite well known in filamentous fungi”. However, we do not say that the mode of regulation that applies to Mr-OPY2 is the major novel finding. How regulation of Mr-OPY2 effects *Metarhizium*’s ability to infect insects is the thrust of this paper.

For the second “possibility”, the reviewer thought, if the Mr-OPY2 is the key finding, this paper is not novel because the reviewer thought the yeast’s OPY2 has similar functions. However, the reviewer’s statement “Mr-OPY2 serves a similar function in *Metarhizium* to the counterpart in yeast” is not correct. It is true that Opy2 has been reported in *Saccharomyces cerevisiae* and *Candida albicans*. But, *S. cerevisiae* is not a pathogen, and its OPY2 is not related to pathogenesis at all. *C. albicans* is a pathogen of mammals. The single paper describing *C. albicans* OPY2 showed that it is dispensable for virulence against its host’s cell lines, but is involved, by an unknown mechanism, in hemocoel infection of a non-natural host (the insect *Galleria mellonella*). In contrast, we found that OPY2 is dispensable for hemocoel infection by the specialized insect pathogen *M. robertsii*. Opy2 proteins have diverse functions in the yeasts themselves, so for example OPY2 is involved *S. cerevisiae*’s tolerance to high osmotic stress, but not *C. albicans*’s. Therefore, Mr-OPY2 is different from its yeast’s counterparts in biological functions, and the transcriptional and translational regulatory mechanisms that apply to Mr-OPY2 have not been reported in yeasts.

We have made significant changes to the discussion section to better illuminate the novelty of our findings.

Major concerns

1. Figure 1. It is not clear what the key (0.7) relates to precisely. The figure legends suggests this relates to the estimated number of amino acid substitutions, but how does this actually relate to the tree. Similarly, the figures in the Table within Figure 1B are not clear to me (statistics, but which?).

Response: The scale bar (0.7), calculated by the Maximum Likelihood provided by MEGA 6.0, corresponds to the estimated number of amino acid substitutions per site, which is stated in the legend. The table in Figure 1B is generated by a standard method for comparing topology of phylogenetic

trees. Nine calculations were used with each giving their statistics for the obtained tree ((A, B), Y) (the presented tree) and the constrained trees ((A,Y),B) and (A, (B, Y)). For example, the obs calculation gives -1.1 for the obtained tree, and 1.1 and 1.6 for two constrained trees, respectively; the smaller the number the better supported the tree is. For other calculations, the higher the number the better supported the tree is. So, all nine calculations consistently show that the obtained tree is better supported than the two constrained trees. This explanation has been added to the legend for clarity.

According to the comment from reviewer #1, this figure is moved to the supplemental section in the revised manuscript.

2. Figure 2B. This is too small to really show differences in appressorium morphology and they are not at all clear from the Figure. No scale bar. Figure 2C also needs a scale bar in both panels. The differences in mycelial colonization in the lower panels are not at all clear and no indication of replication or quantification of this difference is provided.

Response: From this comment to comment #9, the reviewer mostly criticizes the quality of the figures including their size (clarity), legends (not clear because of lack of the information such as replication of the experiments) and scale bars. During preparation of the initial submission, we inserted the figures into the text and submitted all materials in one Word file, which is encouraged by Nature Communications. This substantially reduced the size and clarity of the figures, which we should have noticed and for which we apologize.

Notwithstanding this, for better presentation, we re-prepared figures 1, 2, 3, 4, 5 and 7 (figures 2, 3, 4, 5, 6 and 8 in the first submission) and the new figures are larger and with necessary information such as size markers (DNA, RNA and proteins) and scale bars. Furthermore, we substantially revised the figure legends by adding information such as number of repeats of experiments

3. Figure 3. A No size marker or context at all. No indication of replication of experiment or quantification. Figure 3D No size marker or indication of replication. Figure 3E completely inadequate to show any differences. None of the differences referred to in the text are visible in this figure. No scale bar. Too

small to be of any use, not consistent with text description.

Response: The comments about figures have been addressed in our response above.

4. Figure 4A. No size marker in lower panel. No context, no quantification, no indication of replication of the experiment. Figure 4C. No size marker. No loading control. No evidence of replication.

Response: The comments about figures have been addressed in our response above.

The loading control is moved from the supplement to the main figure.

5. Figure 4D is mentioned in the text. Page 7, line 151 stating that Mr-OPY2 is 1.4 fold higher in hemolymph than in SDY. This panel is not present in the Figure.

Response: Apologies. We were referring to Figure 3D (now Figure 2D in the revised manuscript).

6. Figure 5A. No loading control or size marker (not much use giving this in the supplement). No evidence of reproducibility. Figure 6B. No scale bar. Morphological differences stated in text not clear. Statistical analysis of conidiation differences not provided.

Response: The comments about figures have been addressed in our response above.

Statistical analysis of conidiation differences is provided.

7. Figure 6A needs scale bars. Not really clear at all. The micrograph are no clear and too small. Figure 6C has no loading control or size markers. Figure 6D is not at all clear as a Y2H analysis. Where is dilution and high stringency versus medium/low stringency result. Inadequate control.

Response: The comments about figures have been addressed in our response above.

The loading control is moved from the supplementary file to the main figure.

The yeast two hybrid assays have been repeated with more controls added. This was done under high stringency.

8. Figure 7. The differences in phosphorylation are key to the conclusions of the paper and yet this figure is really very poor. No indication of reproducibility is provided. No quantification of differences is provided apart from the ‘ratios’ which are not explained. This needs to be analyzed properly with replicates, given the critical nature of the data presented.

Response: The comments about figures have been addressed in our response above.

9. Supplemental Figures. I find the ‘YES’ ‘NO’ panels for each of the gels presented to be confusing/inadequate to illustrate the cloning strategies. No size markers on any of the gels shown also compounds the lack of clarity. No size markers provided on any of the protein gels.

Response: the way we presented the pictures for confirmation of gene disruption and complementation has been used in our previous papers (e.g. Zhao et al., PLoS Pathogens, 2014, 10(4):e1004009). Nevertheless, according to the reviewer’s comments, size markers and additional explanation are added for clarity. Additional legends are also added for clarity.

For this reviewers, the lack of rigor in the experimental results presented are a real problem in interpreting the data. I cannot really decide whether the finding is novel and a ‘new insight into fungal lifestyle transitions’ given the poor quality of the data presented. I think they need to evidence the reproducibility of their findings and also provide evidence that Mr-OPY2 is the key gene in this context that they claim.

Response: We did do rigorous experiments with sufficient repeats - this information has now been added to the fig legends and Methods and Materials section.

Four lines of evidences confirm that Mr-OPY2 is a pivotal regulator of insect infection by *M. robertsii*. First, the deletion mutant of *Mr-OPY2* was completely unable to infect insects. Secondly, we demonstrate that Mr-OPY2 protein levels are regulated by alternative transcription start site selection and manipulation of this process changes virulence. Thirdly, RNA-Seq analysis revealed that Mr-OPY2 regulates known virulence genes including cuticle degrading proteases. Finally, the transcription factor AFTF1 is regulated by Mr-OPY2, representing a new circuit regulating fungal infection.

REVIEWERS' COMMENTS:

Reviewer #1 (Remarks to the Author):

I am satisfied with the authors' response and think the ms is acceptable of Nat Comm

Reviewer #2 (Remarks to the Author):

The authors have produced an excellent rebuttal of the comments that I made in the initial review and have corrected the (many) issues associated with the Figures in the original submission. I am now happy with the technical quality of the paper now and the evidence of reproducibility of the findings presented, which was lacking in the original submission, is now made very clear. The figures are, indeed, of a high standard now. I would stress that my criticisms were not cosmetic ones, but rather about reproducibility. There was no evidence of this in the original submission and that was a problem.

In terms of the findings themselves, I am still not completely convinced by the novelty of Mr-OPY2 and the argument made that 'in aggregate' the results constitute a major new insight into pathogenesis of *Metarhizium*. However, what is clear is that there is strong evidence of its role in pathogenesis. Furthermore, the data showing that Mr-OPY2 regulates phosphorylation of Hog1- and Fus3-MAPK and the new circuit regulating fungal infection involving Mr-OPY2 and AFTF1 is interesting.

Point-to-point response to reviewers' comments

REVIEWERS' COMMENTS:

Reviewer #1 (Remarks to the Author):

I am satisfied with the authors' response and think the ms is acceptable of Nat Comm

Response: We are glad to know that the reviewer is satisfied. Thanks a lot for the reviewer's critical comments in the initial review.

Reviewer #2 (Remarks to the Author):

The authors have produced an excellent rebuttal of the comments that I made in the initial review and have corrected the (many) issues associated with the Figures in the original submission. I am now happy with the technical quality of the paper now and the evidence of reproducibility of the findings presented, which was lacking in the original submission, is now made very clear. The figures are, indeed, of a high standard now. I would stress that my criticisms were not cosmetic ones, but rather about reproducibility. There was no evidence of this in the original submission and that was a problem.

In terms of the findings themselves, I am still not completely convinced by the novelty of Mr-OPY2 and the argument made that 'in aggregate' the results constitute a major new insight into pathogenesis of *Metarhizium*. However, what is clear is that there is strong evidence of its role in pathogenesis. Furthermore, the data showing that Mr-OPY2 regulates phosphorylation of Hog1- and Fus3-MAPK and the new circuit regulating fungal infection involving Mr-OPY2 and AFTF1 is interesting.

Response: We are very grateful for the reviewer's critical comments on the quality of the figures and on the discussion about the novelty of the findings. Indeed, according to the comments, we made substantial changes to the discussion section, figures and figure legends, which make this paper much better.